# Damage assessment in Braunsbach 2016: A data collection and analysis for an improved understanding of damaging processes during flash floods

Jonas Laudan[1], Viktor Rözer[2], Tobias Sieg[1/2], Kristin Vogel[1], Annegret H. Thieken[1]

[1]University of Potsdam, Institute of Earth and Environmental Science, Karl-Liebknecht-Strasse 24-25, 14476 Potsdam, Germany

[2]GFZ German Research Centre for Geosciences, Department of Hydrology, Telegrafenberg, 14473 Potsdam, Germany

*Correspondence to:* Jonas Laudan (jlaudan@uni-potsdam.de)

**Abstract.** Flash floods are caused by intense rainfall events and represent an insufficiently understood phenomenon in Germany. As an effect of higher precipitation intensities, flash floods might occur more frequently in future. In combination with changing land use patterns and urbanisation, damage mitigation, insurance and risk management in flash flood prone regions are becoming increasingly important. However, a better understanding of damage caused by flash floods requires ex-post collection of relevant but yet sparsely available information for research. At the end of May in 2016, very high and concentrated rainfall intensities led to severe flash floods in several south German municipalities, of which the small town Braunsbach stood as a prime example for the devastating potential of such events. Eight to ten days after the flash flood event, a damage assessment and data collection was conducted in Braunsbach by investigating about all affected buildings and their surroundings. To record and store the data on site, the open source software bundle "KoBoCollect" was used as an efficient and easy way to gather information. Since the damage driving factors of flash floods are expected to differ from those of riverine flooding, a post-hoc data analysis was performed, aiming to identify the influence of flood processes and building attributes on damage grades, which reflect the extent of structural damage. Data analyses include the application of Random Forest, a Random General Linear Model and multinomial logistic regression as well as the construction of a local impact map to reveal influences on the damage grades. Further, a Spearman's Rho correlation matrix was calculated. The results reveal that the damage driving factors of flash floods differ from those of riverine floods to a certain extent. Especially the exposition of a building in flow direction shows a strong correlation with the damage grade and has a high predictive power within the constructed damage models. Additionally, the results suggest that building material as well as various building aspects such as the existence of a shop window and the surrounding might have an effect on the resulting damage. To verify and confirm the outcomes as well as supporting future mitigation strategies, risk management and planning, more comprehensive and systematic data collection is necessary.

# 1 Introduction

Flooding is a common hazard in Central Europe, resulting in high economic losses (Munich Re, 2016). To promote and tailor local planning, flood risk management policies such as the European Floods Directive (2007/60/EC) set up framework conditions for member states to implement flood risk management on national, regional and local levels. Risk assessments and policy decisions are expected to take different flood types into account (e.g. coastal floods, riverine floods, pluvial floods, flash floods), according to the local circumstances (BMUB, 2007). In Germany for instance, storm surge and river flooding are dominant and were therefore considered as risks with national significance. Due to recent severe riverine flooding in eastern and southern parts of Germany, particularly in August 2002 and June 2013, the flood risk management system and the relevant legislation have been substantially improved, among others, according to the EU Floods Directive (2007/60/EC) and its implementation in the Federal Water Act of 2009 (e.g. Thieken et al., 2016a). Supportingly, a large body of literature exists that addresses the topic of riverine flooding in Germany and its effects as well as demands on people, policy makers and general planning. In this regard, risk assessment strategies and effects of preparedness decisions are presented and extensively discussed with a strong focus on recent major riverine flood events. (e.g. Bubeck et al., 2013; Kienzler et al., 2015; Bracken et al., 2016; Osberghaus et al., 2016; Thieken et al., 2016a; Kundzewicz et al., 2017). However, when implementing the EU Floods Directive in Germany, surface water flooding and flash floods were not considered as significant risks and thus neglected. This assessment is currently questioned due to destructive flash flood events in May and June 2016 that caused damage of EUR 2.6 billion (Munich Re 2017).

Flash floods are defined as rapid flood events as an effect of very intense, timely and concentrated precipitation, which are potentially enhanced by orographic features (Gaume et al., 2009; Borga et al., 2014). According to Gaume et al. (2009), flash floods can be triggered by diverse hydrological and meteorological processes and are, compared to riverine and pluvial flooding, associated with a higher number of fatalities. Whereas pluvial floods are related to urban areas and caused by sewage overflow and surface runoff (Maksimović et al. 2009), flash floods usually occur in mountainous regions, where they can trigger debris flows and/or hyperconcentrated flows. Debris flows and hyperconcentrated flows are characterized by the amount of transported and suspended sediment. With a sediment concentration between 60 and 80 volume percent, the quantity of solid material is often higher for debris flows than for hyperconcentrated flows (Gaume et al., 2009). Both flow types show a variation in grain size distribution and deposition characteristics as well: while debris flows potentially carry large debris, boulders and gravel, hyperconcentrated flows transport finer sediments. (Pierson et al. 1987; Gaume et al., 2009; Totschnig et al., 2011; Hungr et al., 2013; Borga et al., 2014).

Weather extremes in Europe are expected to occur more frequently, leading to strong storms, droughts and heavy precipitation in various regions (Beniston et al., 2007; Murawski et al., 2015; Volosciuk et al., 2016). More intense and concentrated rainfall in Central Europe might increase the hazard of severe flash flood events not only in mountainous regions but uplands as well, affecting regions which were previously not perceived as flood prone. Further, an increased risk due to a change in exposed objects and their vulnerability can be detected, which is mainly influenced by urbanisation,

economic growth as well as changing land use patterns (Thieken et al., 2016b; European Environment Agency, 2017). As an effect, flash floods are progressively perceived as a serious hazard in Central Europe. Yet, the implications on elements at risk are poorly understood and assessing their vulnerability, also in comparison to riverine floods, is challenging.

Vulnerability can be defined as the tendency of elements at risk to suffer negative effects and damage, if affected by a specific hazard (Cardona et al., 2012). Regarding flash floods, vulnerability and risk estimations were already conducted in several studies. For instance, Papathoma-Köhle (2016) pointed out that vulnerability assessments for flash floods or debris flows need to be reviewed and adjusted constantly. In her study, an indicator-based method was used for assessing the vulnerability of elements at risk which are exposed to debris flows in South Tyrol. In this regard, the relevance of building characteristics and location for vulnerability estimations were highlighted. Similarly, Fuchs et al. (2012) conducted a study which describes the vulnerability of elements at risk, based on clusters of similar damage ratios caused by flood events. This spatial approach revealed that higher damage ratios are not only an effect of stronger floods, debris flows or hyperconcentrated flows, but are as well dependent on land-use patterns and the characteristics of the elements at risk such as the type and year of construction. With regard to non-alpine environments, Hlavčová et al. (2016) performed a post-hoc analysis of three strong flash flood events which occurred between 1988 and 2004 in northern Slovakia, focusing on the hydrology as well as hydraulic and topographic properties of the catchment areas. They showed that the modelling of flash flood events goes along with major uncertainties due to the lack of data and overall nonlinear relationship between precipitation, runoff and catchment properties.

Concerning flash floods in Central Europe and particularly non-alpine environments, we are in the early stages to understand specific flood and subsequent damaging processes. Especially the aspects of vulnerability estimations of the elements at risk as well as damage driving factors, flash floods are insufficiently understood. Yet, it can be assumed that damage processes of flash floods differ from those of riverine floods, highlighting the need for elaborate research in this field. Riverine floods commonly emerge on the basis of large catchment areas after long lasting rainfall or snowmelt which leads to high surface and groundwater runoff and relatively slow rising water levels. In contrast to riverine floods, flash floods originate from catchments in which geographical features such as steep slopes and defined channels result in rougher flow dynamics in terms of velocity, sediment transport and discharge (Borga et al., 2014). Here, potential damage on buildings comprises erosion and physical impacts, which, on the other hand, do not seem to be distinct damage patterns in riverine flooding (see Kreibich et al. 2009).

To obtain a better understanding of the damage processes of flash floods as well as of effective mitigation options, a comprehensive damage data base that links process dynamics and intensities with damage and loss is needed, but currently not available. Consequently, we present the flash flood in Braunsbach, a town in the district of Schwäbisch Hall in Baden-Württemberg, Germany, as a case study, having collected and analysed data in order to add to the knowledge on damage caused by flash floods and governing factors.

Intense rainfall at the end of May and beginning of June 2016 over Central Europe led to severe surface water flooding and flash floods, which were partly accompanied by mud and debris. Several municipalities mainly in the south of Germany

were hit, eleven people lost their life; infrastructure and buildings were heavily damaged (GDV, 2016). The insured losses of these events amounted to EUR 1.2 billion (GDV, 2016) and the overall loss was estimated to EUR 2.6 billion (Munich Re 2017), an extraordinary monetary loss caused by flash floods in Germany. The district of Schwäbisch Hall in Baden-Württemberg was particularly affected. Moreover, in the beginning of June 2016 the municipality of Rottal-Inn in southern

Bavaria was hit by flash flooding, triggered by the same weather situation (GDV, 2016).

Especially a small village in Schwäbisch Hall, named Braunsbach, faced a severe flash flood on May 29[th] that caused high damage to buildings and infrastructure. The village of Braunsbach is counting just about 1000 residents, yet due to the devastating character and abruptness of this event, the media attention was high and policymakers were attracted. The monetary losses for the municipality of Braunsbach (~2500 residents) were estimated to EUR 104 million, which is more

than 90% of the estimated EUR 112 million of total damage in Schwäbisch Hall (Landkreis Schwäbisch Hall, 2016). The catchment of the creek primarily responsible for the inundation in May 2016, the "Orlacher Bach", is only about 6 km² in size and characterized by steep slopes, in which the stream descends ~180 m over a distance of 3.1 km. Heavy rainfalls in the catchment area between 18:45 and 20:00 of May 29th resulted in an estimated accumulated precipitation of 60 mm, based on radar data which was recorded 70 km south of Braunsbach. Due to inconsistencies and attenuation effects, the data was

corrected up to 153 mm after the approach of Jacobi and Heistermann (2016), (see Bronstert et al., 2017). The extraordinary rainfall patterns finally led to the severe flash flood, which was accompanied by massive amounts of debris and rubble. Streets along the main runoff channel were blocked by layers of debris, up to a thickness of two to three meters while numerous houses in the area showed severe structural damage. Given the relation of town size, event duration and catchment area, the losses were extremely high. Eventually, this event and similar cases of severe flash flooding in Germany triggered a

reassessment of local risk and revealed that the processes and impacts of flash floods are insufficiently understood (in Germany), also showing that research on and management of this particular flood type need to catch up, particularly in comparison to river floods.

Our research paper follows two major objectives. Using the flash flood in Braunsbach as a case study, it is aimed at identifying, analysing, comparing and discussing factors that govern damage caused by this event, applying different linear

and nonlinear methods. As a second issue, the digital methods used for the ex-post damage data collection in Braunsbach and the creation of this database are presented and discussed to demonstrate accompanying challenges as well as advantages during the field work.

## 2 Methods

Collecting and analysing data of structural and non-structural damage to buildings is valuable for understanding specific

damage processes, helping to design and assess effective mitigation measures and creating damage models, which can be used to estimate potential monetary losses ex-ante. Thus, a digital survey was designed to collect relevant information in

Braunsbach which can be used for detailed post-hoc analysis. The type of recorded information is based on existing literature on flood damage surveys (e.g. Thieken et al., 2005; Schwarz and Maiwald, 2007; Merz et al., 2010; Molinari et al., 2014).

## 2.1 Contents of the survey

The survey as well as the data collection was implemented with "KoBoCollect", a self-explaining and network-based open
source software which was developed by the Harvard Humanitarian Initiative together with the Birmingham and Women's Hospital in 2014 (kobotoolbox.org, 2016). The software is designed for quick and reliable information collection after natural disasters or in humanitarian crises. Open source software, as a method for data collection and gaining of knowledge, is increasingly becoming important within the field of natural hazards (Eckle et al., 2016; Klonner et al., 2016). For instance, OpenStreetMap (OSM) and other volunteered geographic information help to create comprehensive databases of up to date
geospatial data which also can be used for natural risk assessment (Schelhorn et al., 2014; Vaz and Arsanjani, 2015; Yang et al., 2016).

The gathered information in Braunsbach included an estimation of damage grades ranging from D1 (no structural damage, slight nonstructural damage) to D5 (very heavy structural damage, very heavy non-structural damage). For this classification, the scheme developed by Schwarz and Maiwald (2007) was adopted to obtain a consistent database and to ensure
comparability with follow-up studies and with data on riverine flood damage (Table 1). Since monetary losses could not be recorded shortly after the event, this classification scheme further offers options for potential subsequent loss estimations. Additionally recorded information included the GPS coordinates, the address (for internal computational use only), the inundation depth at the building in cm, visible damage caused by debris, visible contamination by oil or sewage, the building material and type, specific precautionary measures at the building, the building usage (residential, commercial, public etc.),
the number of storeys and types of outbuildings, the estimated year of construction, the perceived condition of the building before the event, existing shop windows on the ground floor, the existence of a cellar, the sealing degree of the near surrounding as well as the exposition (of the building) in flow direction. All variables except for the address, inundation depth, storeys and the estimated year of construction were pre-coded with the option to record open answers or NA values, resulting in a nominal-, ordinal- as well as interval-scaled data structure. The complete survey with variable descriptions can
be seen in table 2. A more detailed description of the data set, as well as the anonymised data can be found in Vogel et al. (2017a).

## 2.2 On-site data collection

The on-site damage assessment was carried out between June 7[th] and June 8[th], 2016, i.e. 9 to 10 days after the event. The digital survey was conducted by a team of five researchers who investigated all buildings in Braunsbach affected by the flash
flood, using mobile tablet computers with an integrated GPS function.

Some of the flooding characteristics such as flow velocities, grain size as well as the degree of erosion and amount of suspended material could not reliably be determined in the aftermath of the event. Hence, the exposition of the building was

used as a proxy instead. It is assumed that the degree of exposition can be related to flow velocities, hydrostatical forces and (to a certain extent) to sediment/debris load, which in turn leads to different erosion rates at the buildings' foundation. The exposition in flow direction describes the exposition of building walls, corners or parts to the direction and area of the main runoff channel. In this case, a high exposition means that at least one side of the building was fully exposed to water and

potential debris flows. A medium exposition was assumed when parts of the building were exposed, sheltered buildings are characterized by a low exposition.

A thermographic camera (model Testo 876, 160x120 pixels) was used to validate and to derive the inundation depth in such cases, where a reliable estimation through visible traces and marks was not possible. This was done by detecting the remaining moisture in the walls - caused by inundation - through slight differences in surface temperature. A second

advantage of the thermographic camera was the detection of different building materials, which may be covered externally (i.e. plastered half-timbered houses could still be identified as such, see Vogel et al., 2017b).

**2.3 Post-hoc data analysis**

The data was pre-processed and analysed/prepared in R 3.3.1 and QGIS 2.14.3, using the R packages "randomForest", "randomGLM" and "nnet". Since our study aims to identify and analyse damage driving factors of flash floods, the

following variables and binary coded variable expressions were considered and used as predictor variables for the damage grade:

Building material (binary-coded):
- masonry (selected also in case of unidentifiable building material)
- wood
- half timbered

Precaution (binary-coded):
- Different (building) materials (of cellar and ground floor)
- Higher ground floor
- No structural precaution visible

External forces:
- Inundation depth
- Exposition in flow direction
- Contamination visible (binary-coded)
Resistance parameters:
- (Building) condition before event
- Estimated construction year

Various:

- Shop window present (binary-coded)

- Near surrounding sealed

- Having cellar (binary-coded)

- Outbuilding present (binary-coded)

- Private building usage (binary-coded)

The choice of the variables which were specifically analysed was based on both, own judgement (e.g. if the near surrounding is sealed or if an outbuilding or shop window is present) as well as existing literature. Here, Thieken et al. (2005), Merz et al. (2010) and Maiwald and Schwarz (2015) give an overview of important damage influencing factors in case of (river)

flooding, including building characteristics, precaution measures and contamination.

### 2.3.1 Models and correlation tests

Detecting nonlinear and non-monotonic relationships within recorded data becomes increasingly important with regard to flood loss modelling and associated uncertainties (Kreibich et al., 2016). Consequently, a Random Forest model (RF) (Breiman, 2001) was chosen as a method of analysis due to its potential to display nonlinear relationships between variables.

The Random Generalized Linear Model (RGLM) (Song et al., 2013) was constructed as an alternative model to compare the results of the nonlinear RF to a method which implies linear variable coherences. Both models use the same predictor variables and the "damage grade" as dependent variable (excluding cases where no damage was recorded) in order to identify potential damage driving factors (Figure 1 and 2).

The RF is calculated with a number of 500 trees and 4 random variables per split. The number of trees represents the default

settings of the algorithm. The number of variables per split corresponds to the square root of the total variable count (16 in this case, resulting in 4 random variables per split). The RGLM takes 100 iterations with 50 samples per iteration (bag) and a varying count of variables (2 to 16) per bag. Also the number of iterations within the RGLM represents the default settings. The number of samples per bag was set to 2/3 of the total number of observations in the dataset (73 in this case due to the need of complete observations, resulting in 50 observations selected by bootstrapping). The count of variables per bag is

randomly chosen between 2 and the total count of variables. The variable/feature importance of the RF is given by the Mean Decrease Gini, which describes the loss in model performance when permuting the feature values (Breiman, 2001). A higher Mean Decrease Gini indicates a higher importance of the particular variable for the RF model prediction. The feature importance of the RGLM is expressed through the selection count of a variable for the model prediction. By using feature forward selection, a higher selection count of a particular variable indicates a stronger predictive power within the RGLM

model (Song et al., 2013). The performance of both models is given by the rate of false classifications, based on the out of bag predictions. The relative number of cases which were not recognized as the true class is hereby shown in percent (see section 3.2).

Categories with a nominal variable structure (i.e. building usage, building material and structural precaution measures) exist in a binary format, allowing for basic correlation tests. Thus, the identified feature importance from the models was compared to the results from a Spearman's rank correlation matrix. The Spearman's rank correlation was chosen due to the advantage that this method is suitable to analyse variables with different scales of measurements and indicates the strength of

monotonic relationships. Here, the same variables as in the RF and RGLM models were used (Figure 3). An exhaustive list of variable correlations is attached in the appendix, which is based on 51 complete observations within the dataset.

Furthermore, a multinomial logistic regression was applied to test variable coherences between the damage grades and a local impact indicator (Figure 4), which describes a combination of inundation depth and the buildings' exposition in flow direction (the construction of the local impact indicator is explained in section 2.3.2). By treating the damage grade as a

categorical variable, the multinomial logistic regression model gives probabilities of category affinity, given a specific local impact. In order to obtain more data points per category and to reduce modelling uncertainties, the damage grades 3, 4 and 5 were combined into a single class (compare Table 1). The inherent calculations are based on artificial neural networks (Ripley, 1996; Venables et al., 2002) which again do not require any model-specific assumptions such as linearity.

### 2.3.2 Derivation of an indicator "Local impact"

Maiwald and Schwarz (2015) give an up to date overview of factors which influence structural damage on buildings in case of flooding. Especially the building material, condition (before the event) and the age are important factors related to a building's resistance potential. Factors such as inundation level, flow velocity, fluid density, specific energy and contamination relate to "action" parameters and describe external forces (Maiwald and Schwarz, 2015; Milanesi et al., 2015). Thus, in our study, the inundation depth measured at the building and the building's exposition in flow direction were

combined to create a local impact, which can be seen as a proxy for local flood related impact and hydrostatical forces at the building. Consequently, we chose a combination of these factors where both contribute to equal extents. While the inundation depth has continuous values, which are roughly uniformly distributed between 2 and 360 cm for 88 recorded observations (see Table 2), the exposition in flow direction is recorded in three classes (low, medium, high, see Table 2). To achieve comparable variable ranges, the exposition classes "low", "medium" and "high" are transformed into the mean

values of the lower (29 observations), middle (30 observations) and upper third (29 observarions) of recorded water levels. The derived values 56, 135 and 232 fit into the range of observed water levels, enabling a combination of both attributes (Figure 5). The calculated local impact corresponds to the sum of water level and transformed exposition value. Please note that the exposition values are not used to replace water levels, but are only transformed into a comparable range.

Furthermore, a local impact map was created in QGIS (Figure 6) by calculating Voronoi diagrams for the geocoded data

points and solely displaying the area with affected houses. For simpler visual appearance and better distinction of the displayed data, the Voronoi diagrams were smoothed by a Gaussian filter. The local impact map is used for visualization and comparison of the local impact indicator to the spatial distribution of the damage grades, since potential areas of similar local

impact between and around the buildings are shown. However, it has to be noted that the local impact was measured directly at the buildings and is therefore hypothetical for the areas around.

## 3 Results and discussion

The flash flood in Braunsbach was accompanied by a considerable amount of sediment and building rubble, potentially
showing flow characteristics of debris flows such as defined by Totschnig et al. (2011), Hungr et al. (2013) and Borga et al. (2014). Yet, a clear distinction between flash floods and debris flows is not always straightforward and could not be reliably determined in the field. Throughout our discussion, we will therefore use the term "flash flood" only. The following section begins with a general reflection on the data collection process, limitations and data quality. Thereafter, the damage influencing factors are identified and discussed by applying different linear and nonlinear methods. Finally, features such as
the local impact and the damage grades are spatially visualised, helping to discuss our outcomes and illustrating the flash flood processes in Braunsbach.

### 3.1 Data collection and field work; assumptions and limitations

The in-field work load can be estimated to roughly 10 hours, in which a team of five researchers was able to survey 96 buildings in Braunsbach (corrected to 94 observations after database checks), for each specifying 18 variables. In addition, a
picture has been taken along with the coordinates, the address and if needed, further details regarding the building's usage. Table 2 provides an overview of the data types and frequency distributions. One week after the event, the structural damage on buildings and building characteristics were still assessable, since the main work within this period was focused on clearing the roads, establishing paths for large construction machinery as well as removing and cleaning the interior of affected buildings. The progress of the clean-up work was even beneficial for the damage assessment to a certain degree, as a
thick layer of debris and rubble previously covered big parts of the damaged buildings. However, a few buildings could not be reliably examined, since debris and rubble were still considerably hampering the access.
When handling the thermographic camera it has to be pointed out that, even one week after the event, remaining moisture and visible traces could still be detected without problems. Yet, ascending humidity in the walls is a point to consider when using a thermographic camera for water level estimations. Rising moisture can distort the observation of actual water levels
at the building. For that reason the thermal images were checked against estimations based on visible mud contamination and marks caused by water and transported debris as well. Since the thermally derived water levels matched well with visible traces, the inundation depth for buildings derived from thermal images could be accepted without any correction. Still, when using a thermographic camera for water level estimations on buildings, it has to be considered that the type of flood (flash flood, riverine flood) has an effect on the duration of the inundation and thus on the distinctness of visible moisture
boundaries. Considering the short inundation times in Braunsbach, the overall good visibility of moisture boundaries was remarkable.

Overall, the in-field data collection was greatly facilitated by the use of "KoBoCollect" in terms of speed, handling of the gathered data and efficiency of data processing and analysis. However, to create a uniform database and to maintain consistency among the different team members throughout the data collection process, objective criteria for items such as the structural damage had to be defined. Therefore, careful preparations and agreements were carried out prior to the field trip off site as well as on site. In retrospect, we consider the data to be consistent in a way that the team members had very similar opinions e.g. on the damage grades or exposition in flow direction. Thus, a bias in the dataset due to personal variations in expert judgement is expected to be low. This assumption is further supported by the engineering analysis of Maiwald and Schwarz (2016), who applied the same damage classification system to assess the buildings structural damage in Braunsbach. Their report reveals that the distribution of the recorded damage grades after a second inspection (D1: 40, D2: 43, D3: 5, D4: 7, D5: 3) is relatively similar to the distribution presented in this study (D1: 39, D2: 34, D3: 5, D4: 6, D5: 5 as shown in Table 2). Although it is not known which damage grade was assigned to which building, it is likely that, even among people with different qualifications (experienced engineer, researcher or student), comparable results can be achieved and that data collection can be consistent. This offers interesting options for crowdsourced information collection using open source software such as "KoBoCollect", which can be helpful for scientific research.

## 3.2 Models and Correlation tests

First, the collected data were used to identify damage driving factors by creating a Random Forest model (RF) and a Random Generalized Linear Model (RGLM) with the damage grade as response variable. In a next step, a Spearman's rank correlation matrix was constructed. In the following, the different model and correlation results are discussed and compared to each other.

The post-hoc data analysis revealed that the RF and RGLM both show a relatively poor model performance, based on the false classifications. Here, the percentages of false classifications for the RF are 33.3% for damage grade D1, 41.9% for D2 and 100% for D3 and higher. The RGLM performed slightly better with a false classification of 33.3% for D1 and 41.9% for D2 as well, 20% for D3, 80% for D4 and 100% for D5. However, trends and relations of predictor variables with the damage grade can be derived. Both models give the highest feature importance for the damage grade to the inundation depth and the exposition (of the building) in flow direction. Here, the Mean Decrease Gini for the RF was 11.3 and 6.9 (average: 2.7), whereas the RGLM feature selection count in 100 iterations was 96 and 89, respectively (average: 39) (Figures 1 and 2). It is further shown that the RGLM compared to the RF indicates a different variable importance hierarchy for variables other than the inundation depth, the exposition in flow direction and the estimated construction year. This is due to different internal calculations of the variable importance, as explained in section 2.3.1. Yet, this issue also suggests that, apart from the inundation depth, the exposition in flow direction and possibly the estimated construction year, differences in variable importance are less distinct in both models and the predictive power is low which hampers the interpretation of the importance hierarchy when comparing both models.

Regarding the correlation tests, the highest positive (and significant) correlations can be seen between the damage class and the exposition of the building in flow direction as well as the damage class and inundation depth with a value of 0.69 and 0.66, respectively (Figure 3). Hence the correlation analysis strongly confirms the results of the RF and RGLM. The detected strong link of the exposition of the building in flow direction and the inundation depth to the caused damage makes sense,

given the nature of the event and the mass of debris, water and mud flowing down the main channels within the village of Braunsbach. These results are confirmed by Maiwald and Schwarz (2016) as well, who identified the exposition of a building to the flow direction as an important parameter for potential structural damage. A high exposition in flow direction can be related to a higher flow force of water, higher flow velocities and intensities acting on a building. Investigations on these parameters regarding riverine floods by Kreibich et al. (2009) resulted in weak correlations with recorded damage

grades of residential buildings. At this, it is revealed that especially the exposition in flow direction is a significant damage driving factor of flash floods, which does not show strong importance in riverine flooding.

The estimated construction year of a building displays a certain importance within the RF as well as the RGLM model with a Mean Decrease Gini of 5.9 and a feature selection count of 52, respectively (see Figures 1 and 2). In this case, the correlation analysis does not reveal any significant monotonic relationships between the estimated construction year and the damage

grade (Figure 3). Additionally, the building condition before the event displays only slight importance within the RF and slight, non-significant correlation with the damage grade. Since the construction year is related to the overall preservation and the building's state of the art in terms of technology, it can still be assumed that newer buildings or buildings in a better condition have a higher resistance to structural damage. This is in line with Maiwald and Schwarz (2015) who consider the building age and condition to have an influence on the expected structural damage.

Further, a positive correlation, but of low significance, can be observed between the damage grade and the existence of a shop window with a value of 0.19 and a p-value of 0.11 (Figure 3). Accordingly, the RF model shows a certain variable importance with a Mean Decrease Gini of 2.4 (Figure 1). Here, a trend towards higher damage grades caused by shop windows on ground level which - in case of breaking - open debris and water paths to the inside of the building can be assumed. Also Maiwald and Schwarz (2016) underline the fact that broken windows may allow water and debris to

accumulate inside the building, causing damage to sustaining building structures. Yet, our results might also be affected by the fact that in this case study, buildings with shop windows mainly occur along the main street and city centre and are therefore located inside the main flow channels.

No (obvious) precaution at the property level indicates a slightly higher chance for higher damage as well by displaying a positive correlation of 0.1 with the damage class, although the significance is low (p-value 0.4) and there is no remarkable

importance within the RF and the RGLM models. Yet, this is in line with the negative correlation between the damage class and the precaution measure "different (building) materials (of cellar and ground floor)" of -0.19 which in fact shows a low significance (p-value 0.11) but still allows for meaningful assumptions. This is supported by Thieken et al. (2005) and Merz et al. (2010) who claim that different precautionary measures significantly reduce the damage on buildings in case of

flooding. Still, the question arises to which degree precautionary measures, which were effective at riverine flooding, are suitable to lower or mitigate structural damage on buildings in case of flash flooding.

The building material masonry seems to have a slight damage reducing effect by displaying a negative correlation of -0.09 (p-value 0.43) with the damage class while the building material half-timbered shows a very slight but non-significant

positive correlation of 0.08 (p-value 0.49) with the damage. Interestingly, the RGLM model only considers the building material masonry as relatively important for the damage grade prediction, whereas the RF does not display a significant feature importance for all building materials. Although it is not clearly shown which distinct building material is related to lower structural damage, it can be assumed that if being hit by debris, half-timbered houses are more susceptible to structural damage than houses made of masonry and concrete due to their lower structural stability (Schwarz and Maiwald, 2007).

Overall, when performing detailed analyses such as models and correlation tests it has to be considered that the database of 94 data points is rather small and assumingly insufficient for creating representative and universal results. This fact could also explain the low model performances and low significances in some of the cases discussed above. Nonetheless, it is important to point out the strong correlations in many cases of up to 0.69 (damage class and exposition in flow direction) revealing obvious damage driving factors and showing as well that the data collection within the team of different

researchers was consistent.

### 3.3 Evaluation of the local impact

In Figure 6, the town of Braunsbach as well as the corresponding local impact during the event and recorded damage grades are illustrated. The map reveals that highly damaged buildings and a strong local impact - which relates to hydrostatical and impact forces at the building (see section 2.3.2) - occurred along the main runoff channels of water and debris during the

event. Higher damage grades were also recorded in the lower-lying town regions, where the tributaries "Orlacher Bach" and "Schlossbach" flow into the river "Kocher", since debris and water accumulated in these areas and caused severe structural damage. Most of the higher damage grades are located in high local impact areas. Yet, especially in those areas, the degree of damage differs strongly, highlighting the complexity of damage driving processes that cannot be explained by the local impact alone. The flow characteristics of debris and rubble during severe flash floods can be unforeseen and influenced by

chaotic factors, changing sediment deposition as well as bedload processes (Totschnig et al., 2011; Hungr et al., 2013). Thus, it can be assumed that, during the flash flood in Braunsbach, chaotic factors and deposition of debris led to various damage patterns which remain inexplicable through quantitative analysis and modelling. This is strongly supported by the engineering report of Maiwald and Schwarz (2016), who claim that chaotic flow processes at the building caused by rubble and debris can greatly influence the inundation depth.

However, it is revealed that buildings with a high exposition in flow direction are more susceptible to severe structural damage, since the probability of large debris colliding with building walls is much higher and erosion of the foundation is more likely to happen. Also Maiwald and Schwarz (2016) stated that the recorded damage patterns differ from damage patterns caused by riverine flooding and appear to be more severe due to higher hydrodynamic stress and collision of debris

with the building. Conversely, some buildings can benefit from shadowing effects of neighbouring buildings, which retain debris and suspended material to a certain degree.

To further evaluate the local impact indicator, a multinomial logistic regression was applied. By analysing the dependency between only the local impact indicator and the damage grade, the influence of external forces on the damage grade can be observed separately, since resistance parameters and building characteristics are neglected. As can be seen in figure 4, there is a clear coherence of an increasing local impact and an increasing probability to belong to higher damage grades. However, in accordance with figure 6, figure 4 reveals again that external forces are not enough to explain the complex damage pattern. Especially for moderate impact values (around 300 to 400) non-negligible probabilities are assigned to all damage grades. Further, if higher local impact values are considered, a large model uncertainty has to be taken into account, which is shown by the 95% confidence interval that covers a probability range of 45% for the corresponding damage grade affinity. This means that i.e. given a local impact of 550, the chance of belonging to class D2 ranges from 5% to 50% and for greater equal D3 from 50% to 95%. The large variability can be explained by the small number of observed data points with high local impact values. Additionally an increasing complexity of the damaging process for higher local impact values might contribute to the model uncertainty. Still, figure 4 shows that a local impact indicator can be suitable to evaluate the hydrostatical forces of this type of hazard, which, in addition to the characteristics of the element at risk, might allow vulnerability estimations such as performed by Papathoma-Köhle (2016).

It can be summarised that, next to individual flow- and deposition processes of the flood, local factors, shadowing processes, and building characteristics shared a certain importance as damage driving factors in Braunsbach, highlighting the complexity of this event. This is supported by the findings of Fuchs et al. (2012) who revealed as well that damage on buildings is not only caused by flood-inherent processes and intensities, but is also influenced by building characteristics and dependent on the general land use pattern.

According to Varnes (1984), risk reflects the expected damage which is governed by the hazard, exposure and vulnerability. Our results show that the local impact - which stands as a proxy for elements of the hazard processes and the exposure - is a meaningful external indicator for structural damage caused by flash floods although it does not fully explain the recorded damage grades. It can be used either in multi-variable damage models or in future risk maps for flash flood prone regions, introducing a valuable parameter for current and following risk and damage assessments. However, questions arise on how to collect necessary data for a reliable calculation of respective values. A feasible option is the derivation of values from aerial images in combination with digital elevation models to identify buildings which are exposed or shielded. Given the specific type of hazard, in this case flash floods, a local impact according to potential inundation depths and a building's exposition in flow direction could be estimated either manually or by algorithms. Prerequisites and challenges however comprise the accessibility of data, up-to-dateness, adequate image resolutions and quality checks. Here, further research is needed to evaluate potential uses of indicators such as the local impact, which can be relatively easy derived and hold a proxy character.

An alternative and quantitative approach to assess hydraulic forces on buildings is the computation of flow fields during flash floods, taking into account local slope and fluid densities. This approach is presented by Milanesi et al. (2015), who introduce a conceptual model which describes the acting forces on humans during rapid floods. However, detailed information about the buildings shape and geometry, friction coefficients as well as flow dynamics are required for the computation.

Consequently, when performing damage and risk assessments for flash floods in future, compromises must be found on issues such as the robustness and uncertainties of models, data availability as well as efficient data handling.

## 4 Conclusion

The evaluation and data analysis in this study resulted in important information about the impacts (damage to buildings) of the flash flood event in Braunsbach. It is revealed that not only the water depth, which is often considered as only damage driving factor in riverine flood loss modelling, but also the exposition of a building in flow direction and susceptible building parts like e.g. shop windows seem to be risk factors in flash-flood prone regions. This result considerably differs from investigations on damage caused by riverine floods. Yet, the damage driving as well as damage reducing factors of flash floods are complex, often unpredictable, contingent upon the surrounding as well as dependent on certain building characteristics.

Knowing processes of flash floods and their impacts can help to create awareness for future events and support strategic planning with regard to similar emergencies. Concerning the European Floods Directive 2007/60/ EC and its implementation in Germany, implications according to the German Federal Water Act exist. The consideration of flash floods and surface water flooding as a "significant risk" would result in the obligation to create new nationwide hazard and risk maps. As a further consequence, the German Federal Water Act intends a building ban in all areas that are affected by a 100-year flood event, which would lead to serious consequences for local planning in flash flood prone regions. Therefore, flash floods are currently judged as a "general risk" throughout Germany.

Still, maps such as the presented local impact map could be a supportive and feasible first step in order to update and perform risk and damage assessments. The estimation of a local impact could be used in integrated risk management and strategic planning of mitigation measures against future hazards in Braunsbach or similar villages in that region. Thus the introduced concept may be beneficial for the identification of potentially vulnerable locations on a small scale and within case studies, helping to understand the potential future development of flash flood prone regions. However, further investigations are needed in order to verify the results and to obtain larger databases.

To facilitate data collection in the future, the case further demonstrates the potential of mobile devices and open-source applications. In the field, the simplicity, speed, quality and handling of information using the open source application "KoBoCollect" particularly stood out as a great advantage. Even in a short time and with a small team of researchers it was possible to gather a fair amount of useful information that could be further processed and analysed. The public availability of

the software makes it a fast and ad-hoc tool for assessing different kind of questions, usable in various research fields and not only for scientific but also for private uses. However, further aspects to discuss are whether the quality of crowdsourced information is suitable for scientific investigations and how to approach and deal with possible limitations, security and copyright issues as well as uncertainties. Still, it can be concluded that open source data collection software for mobile use

has great potential as a scientific tool to generate extensive valuable data under challenging conditions. It should be especially considered in time critical research applications such as ex-post-disaster analyses, as was demonstrated by the presented case of Braunsbach.

## Competing interests

The authors declare that they have no conflict of interest.

## Acknowledgements

The presented work was developed within the framework of the Research Training Group "Natural Hazards and Risks in a Changing World" (NatRiskChange) funded by the Deutsche Forschungsgemeinschaft (DFG; GRK 2043/1). We further appreciate the help of Benjamin Winter, working at the alpS GmbH in Innsbruck, Austria, as well as giving thanks to Melanie Eckle, Benjamin Herfort, Carolin Klonner and Chiao-Ling Kuo from the University of Heidelberg for supporting

the damage assessment in Braunsbach and providing additional mobile tablets for the data acquisition. Special thanks to the contribution of Viktor Rözer, which was partly funded by the German Ministry of Education and Research (BMBF, 03G0846B) via the project "EVUS—Real-Time Prediction of Pluvial Floods and Induced Water Contamination in Urban Areas".

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

**Table 1: Assignment of damage grades $D_i$ to damage cases; examples from the flood in August 2002 (after Schwarz and Maiwald 2007)**

| Damage grade | Damage pattern (sketch) |
| --- | --- |
| **D1:** no structural damage, slight non-structural damage<br><br>   -   moisture penetration of walls and ceilings |  |
| **D2:** no structural damage to slight structural damage, moderate non-structural damage<br><br>   -   moisture penetration and contamination<br>   -   small cracks in walls, dented doors and windows |  |
| **D3:** moderate structural damage, heavy non-structural damage<br><br>   -   larger cracks and in walls, dented doors and windows<br>   -   beginning subsidence of the building<br>   -   replacement of building components necessary |  |
| **D4:** heavy structural damage, very heavy non-structural damage<br><br>   -   collapse of load-bearing walls, large cracks<br>   -   replacement of load-bearing components necessary |  |
| **D5:** very heavy structural damage, very heavy non-structural damage<br><br>   -   collapse of large building parts<br>   -   demolition necessary |  |

**Table 2: Features of 94 buildings affected by flooding in Braunsbach, Germany, recorded between 6[th] and 9[th] June 2016, and their frequency of occurrence.**

| Variable | Characteristics | n |
|---|---|---|
| Damage grade | D1 (no structural damage, slight non-structural damage) | 39 |
| | D2 (no to slight structural damage, moderate non-structural damage) | 34 |
| | D3 (moderate structural damage, heavy non-structural damage) | 5 |
| | D4 (heavy structural damage, very heavy non-structural damage) | 6 |
| | D5 (very heavy structural damage, very heavy non-structural damage) | 5 |
| | No damage | 5 |
| | NA | 0 |
| Inundation depth (cm) | "Integer value" | 88 |
| | NA | 6 |
| House type | Single-family house | 46 |
| | Apartment building | 25 |
| | Semi-detached house | 3 |
| | Terraced house | 0 |
| | NA | 20 |
| Building material | Masonry | 71 |
| | Half-timbered | 26 |
| | Wood | 10 |
| | Concrete | 0 |
| | Steel | 0 |
| | Rubber | 0 |
| | NA | 1 |
| Building usage | Residential | 58 |
| | Commercial | 8 |
| | Combined/Mixed | 21 |
| | Public services | 6 |
| | NA | 1 |
| Near surrounding sealed | Yes | 64 |
| | Mainly yes (small areas around not sealed) | 21 |
| | Mainly no (larger areas around not sealed) | 8 |
| | No | 0 |
| | NA | 1 |
| Exposition in flow direction | High (at least one side of the building fully exposed to water flow) | 34 |
| | Medium (parts of the building exposed to water flow) | 34 |
| | Low (sheltered by other buildings / slightly exposed to water flow) | 26 |
| | NA | 0 |
| Damage caused by debris | Yes | 55 |
| | No | 37 |

| | NA | 2 |
|---|---|---|
| Building condition before event | Good | 45 |
| | Medium | 46 |
| | Bad | 1 |
| | NA | 2 |
| Outbuildings present | Yes | 32 |
| | No | 59 |
| | NA | 3 |
| Type of outbuilding | Garage | 11 |
| | Carport | 1 |
| | Barn | 8 |
| | Shed | 7 |
| | Summerhouse | 1 |
| | Greenhouse | 0 |
| | Conservatory | 0 |
| | Other | 7 |
| Number of storeys | "Integer value" | 93 |
| | NA | 1 |
| Shop window | Yes | 18 |
| | No | 74 |
| | NA | 2 |
| Having cellar | Yes | 30 |
| | No | 57 |
| | NA | 7 |
| Estimated construction year | "Integer value" | 88 |
| | NA | 6 |
| Structural precaution | Higher ground floor | 19 |
| | Different (building) materials (of cellar and ground floor) | 23 |
| | Protection of cellar duct | 3 |
| | Other | 4 |
| | No precaution | 52 |
| | NA | 3 |
| Contamination visible | Yes | 77 |
| | No | 15 |
| | NA | 2 |
| Contamination type | Oil | 4 |
| | Chemicals | 0 |
| | Sewage | 0 |
| | Mud | 77 |
| | Other | 0 |

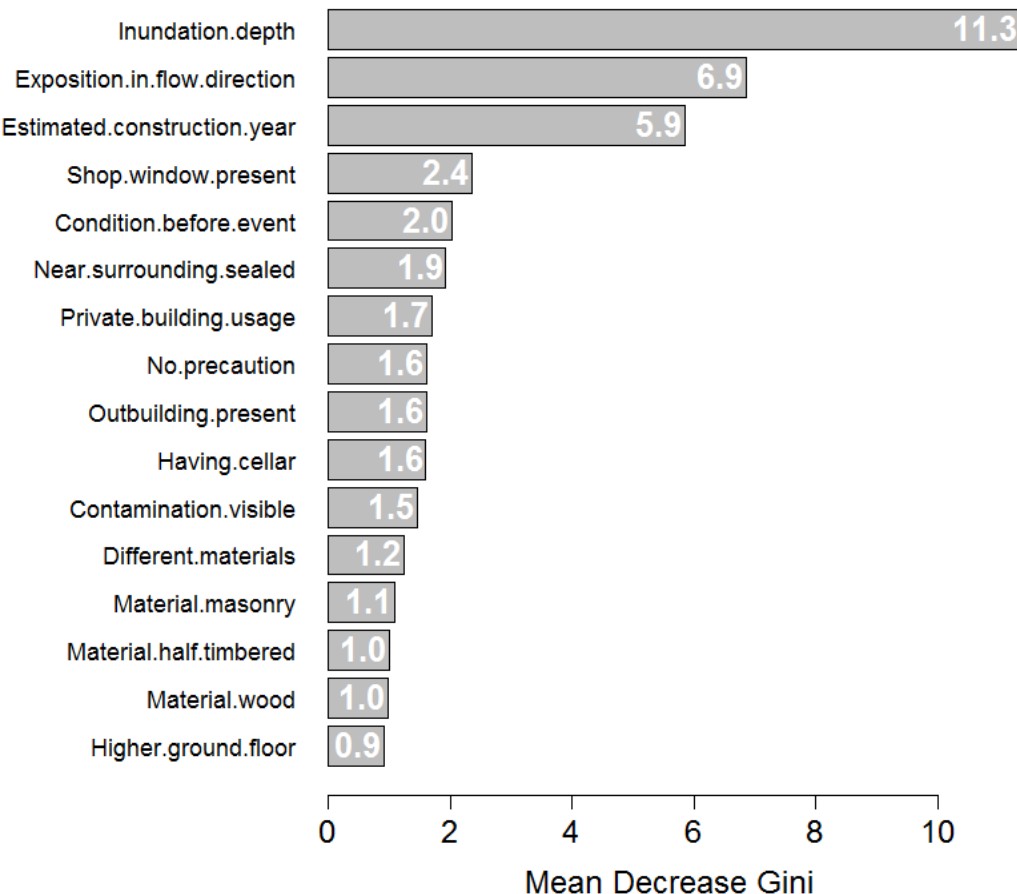

**Figure 1: Random Forest feature importance (Mean Decrease Gini) for the response variable damage grade.**

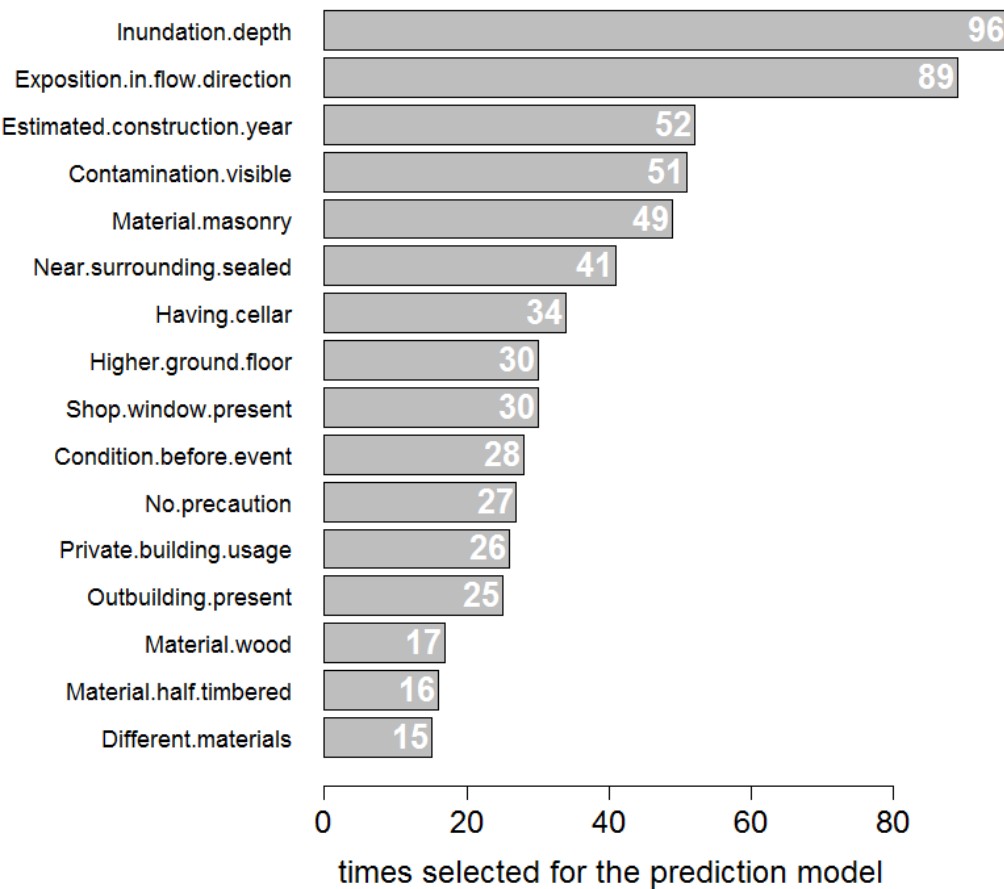

**Figure 2: Random Generalized Linear Model feature importance (times selected) for the response variable damage grade..**

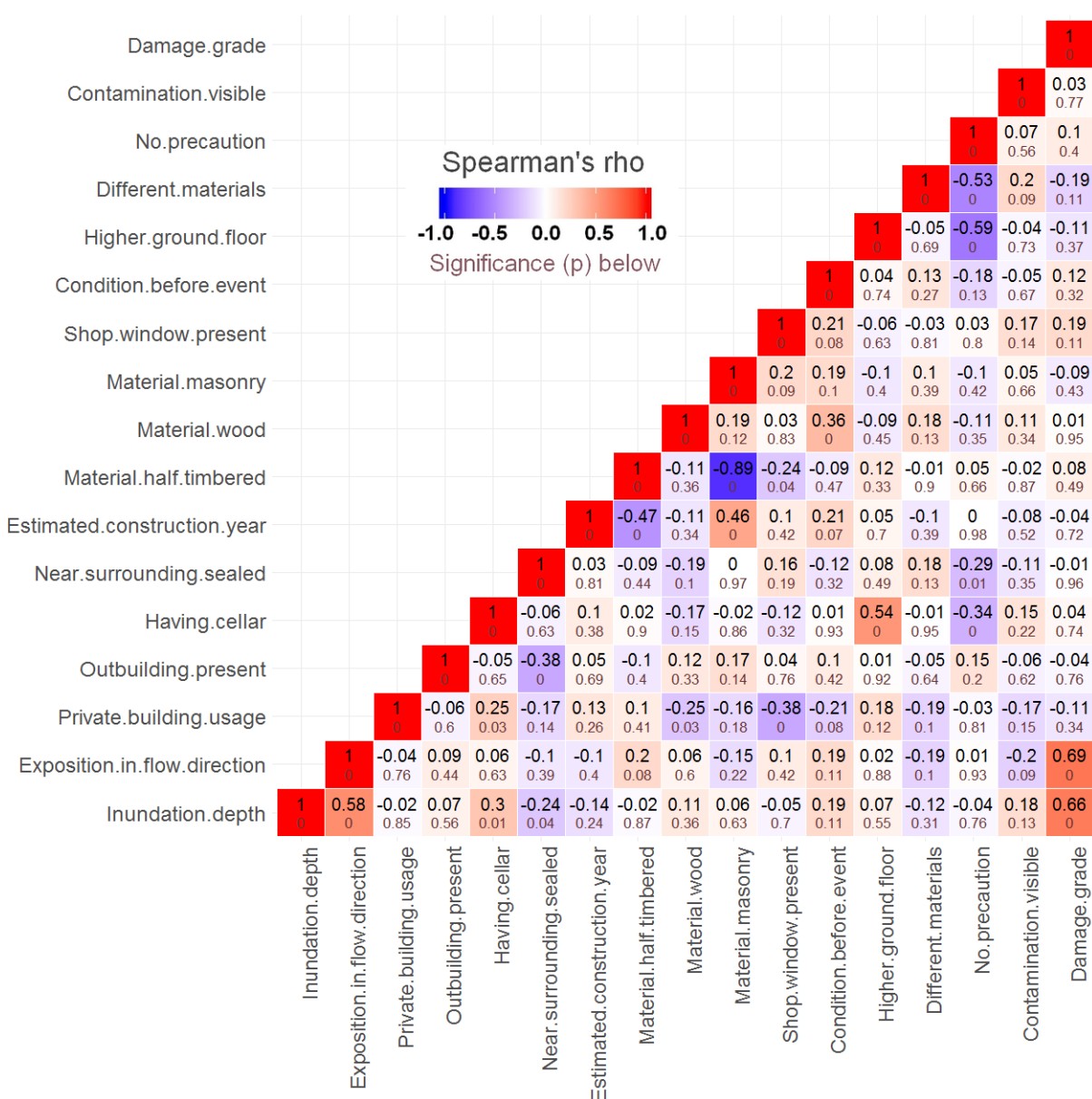

**Figure 3: Spearman's rank correlation matrix and correlation significances of relevant variables (see Table 2 for a description of the variables). The count of complete cases for the analysis was 73.**

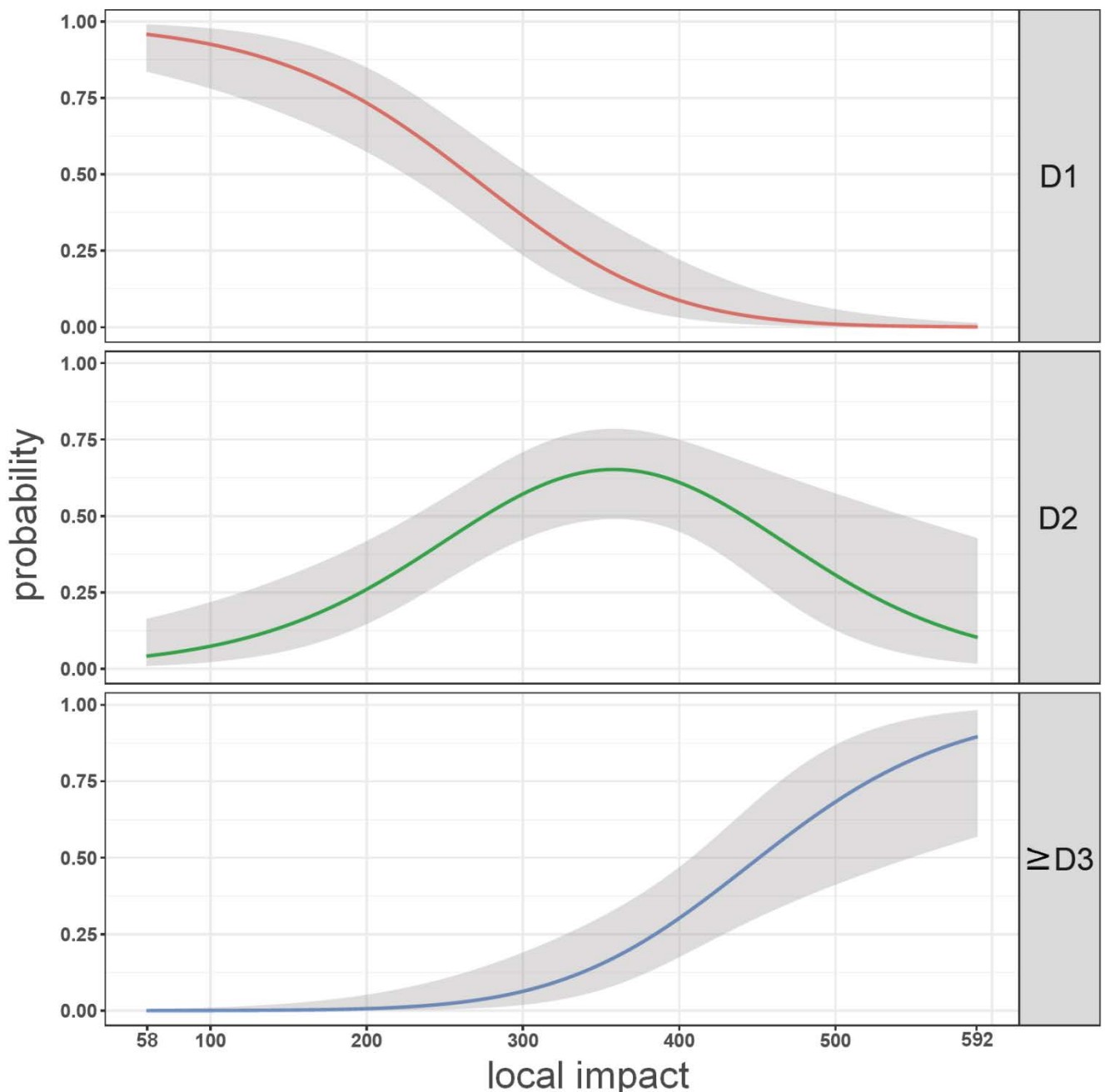

**Figure 4: Probabilities of the damage grade predicted by the multinomial logistic regression model (see Table 2 and section 2.3.2 for details on the damage grades and the local impact indicator). It can be seen how the probability for a specific group affinity changes with an increasing local impact value and shifts towards higher damage grades.**

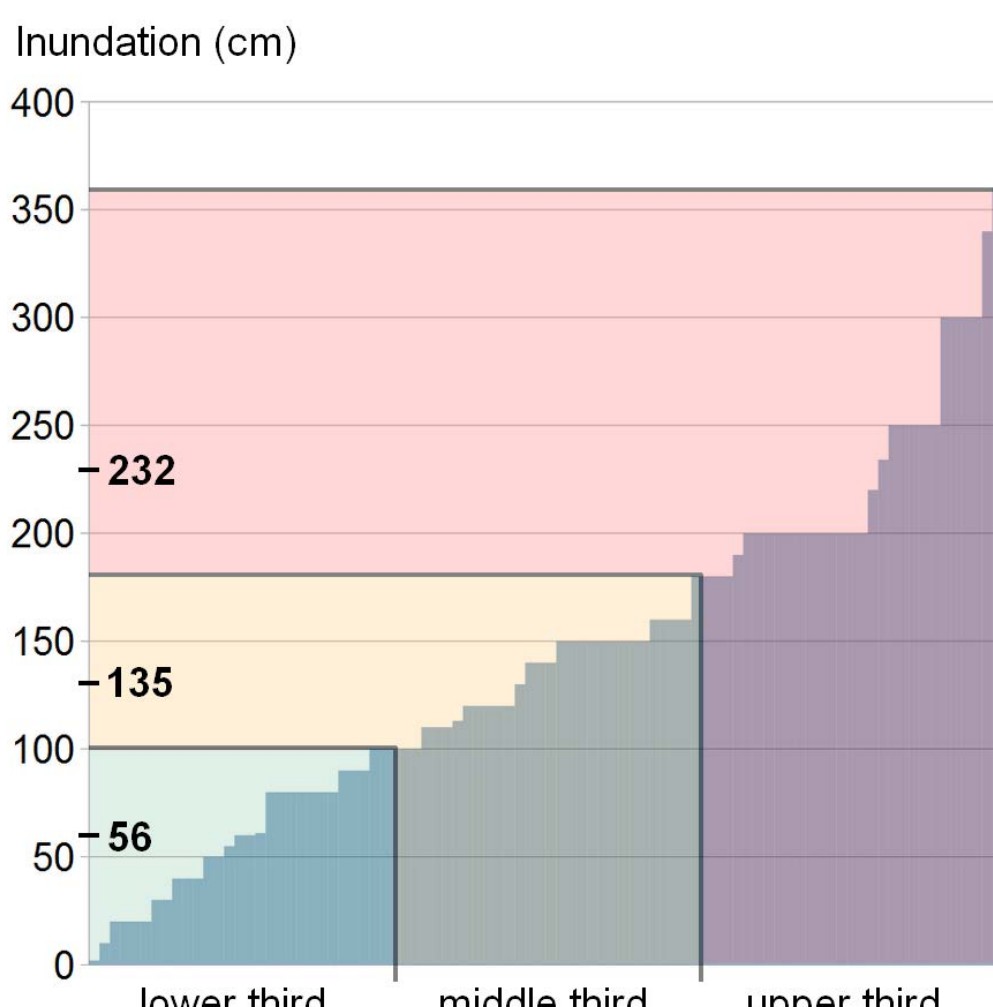

**Figure 5: Step of deriving the local impact indicator. The recorded inundation depth is sorted in ascending order. By sorting, the relatively uniform distribution of the inundation values is shown, which allows the general procedure. On the left, the mean values of the lower, middle, and upper third of the sorted inundation depth (56, 135, 232) are given, which were used to replace the exposition classes "low" "medium" and "high". This step enables a comparable variable range and the derivation of an interval-scaled indicator for further analysis.**

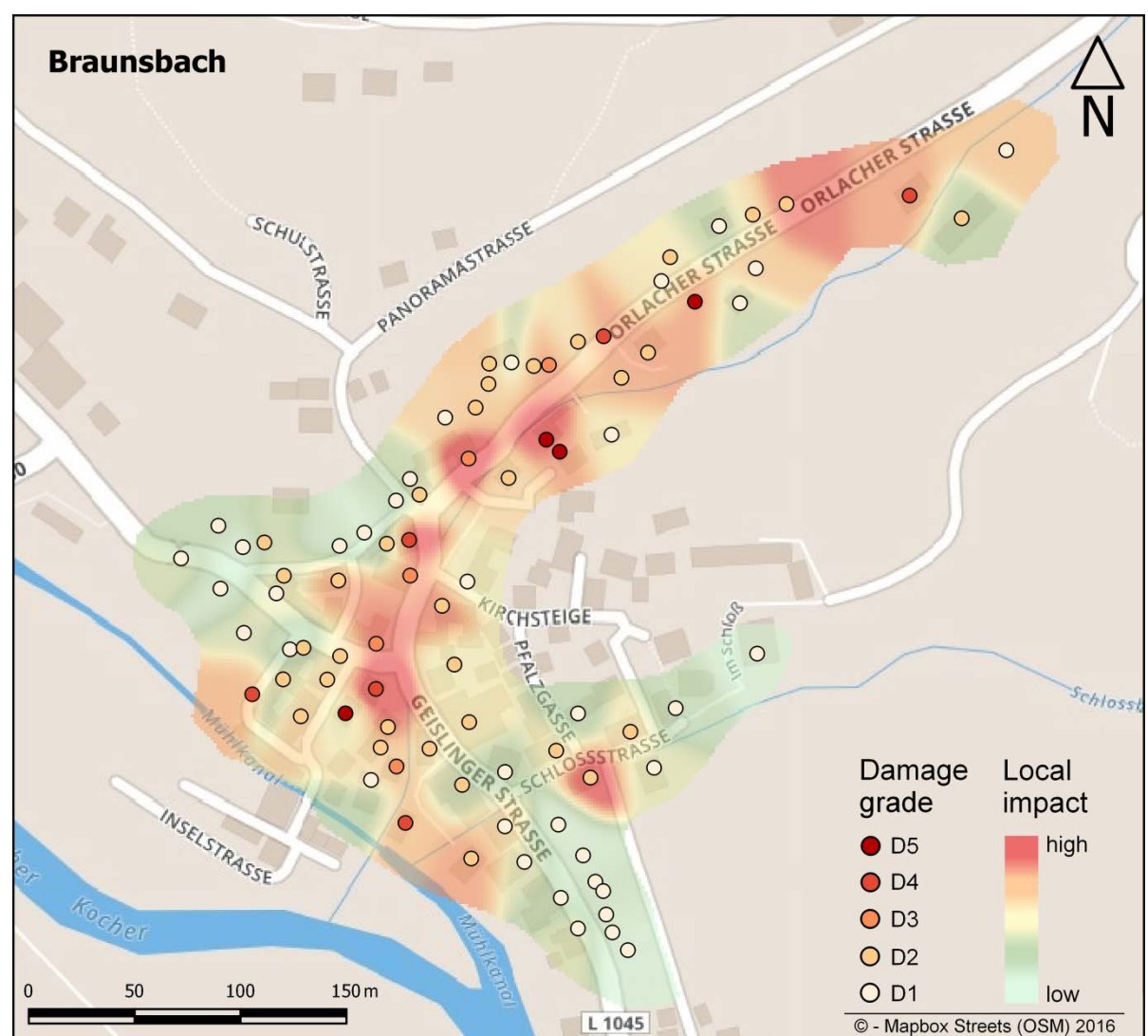

**Figure 6: Map of the study area with the local impact, which is a combination of the inundation depth at the building and its exposition in flow direction (see text for further details). Further, the damage grades as recorded on site using the classification scheme of Schwarz and Maiwald (2007) are shown; see Table 1 for a verbal description of the damage grades.**

# Appendix

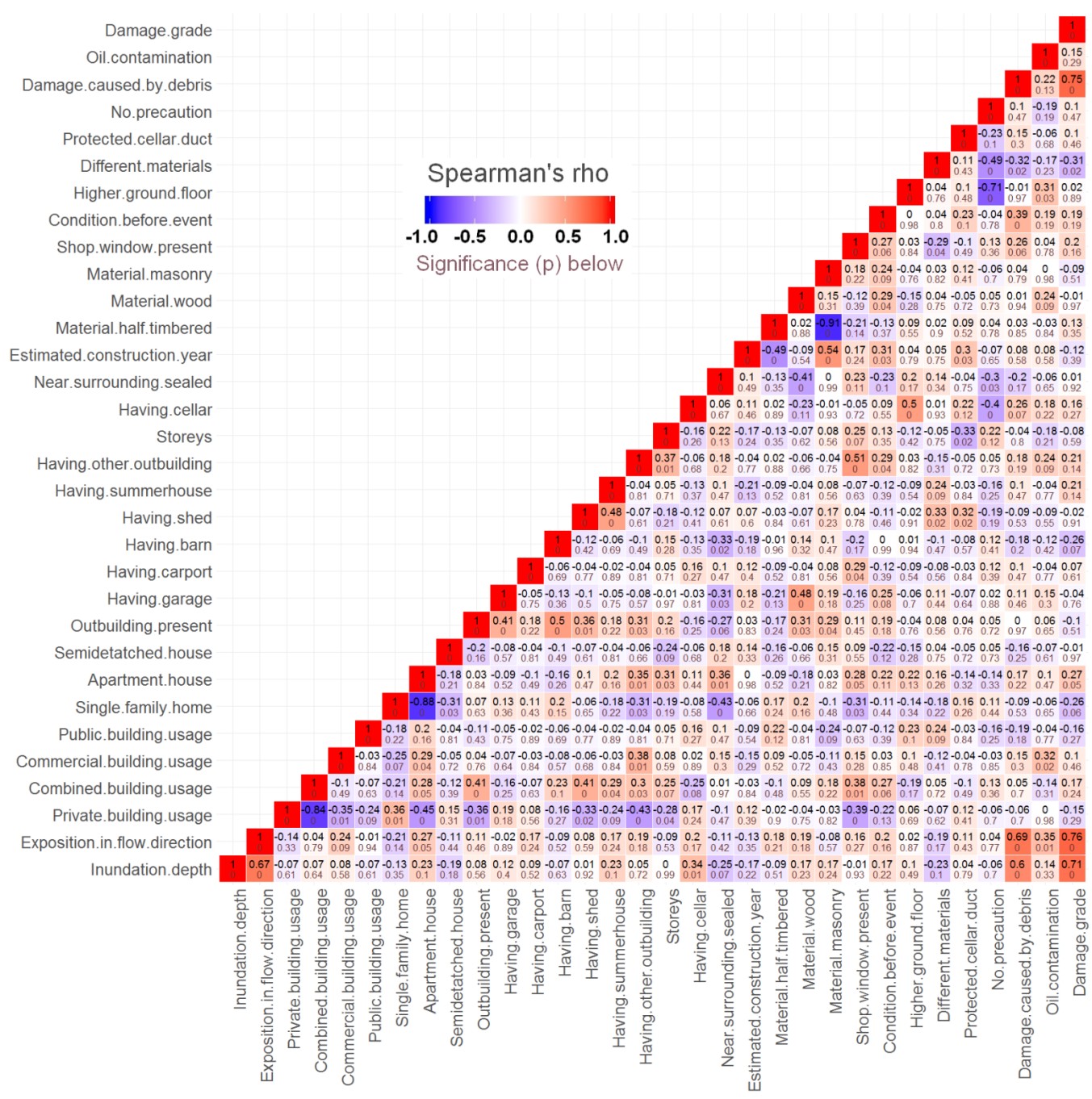