# Peer review of "Damage assessment in Braunsbach 2016: A data collection and analysis for an improved understanding of damaging processes during flash floods"

_Natural Hazards and Earth System Sciences, 2016_

## Referee Comment (RC1) · Anonymous Referee #1 · 10 Jan 2017

I have read the brief communication "Brief Communication: on site data collection of damage caused by flash floods: experiences from Braunsbach, Germany, in May/June 2016" with great interest. I would not recommend the publication of the present work as brief communication but the resubmission after major revisions as a Research paper.

My major concern is that the work presented here is not appropriate for a brief communication. To my knowledge a brief communication should present ground breaking and new results, findings, methods and/or observations usually following an event (in this case the flash flood of Braunsbach) that needs to be communicated to the scientific community as soon as possible (e.g. lessons learnt). In my opinion, the present work should be written and presented as a research paper. In that case, the authors

should include a short literature review citing similar work that has been done (not only in the field of floods) as well as existing damage data collection methods but also they should round up their work by explaining how the results should be used. They have to be more precise in describing their motivation to conduct this work and their clear aims. They also need to invest more time and text in explaining why they chose the specific methods (t-test, Spearman's rank correlation matrix) but also the specific variables. Either way, the proposed revisions should address the following points: General comments: 1. It is not very clear for which process the research has been conducted for. The title refers to "flash floods", however, in the text the term "debris flow" is often used for the process under investigation (e.g. p. 2, line 31). Are these two processes identical for the authors? What is the difference of these processes regarding their impact on buildings? Were all the buildings under investigation impacted by the same process?

2. The article should refer to similar studies and their connection to them, for example:

Papathoma-Köhle M., Zischg A., Fuchs, S. Glade T., Keiler M. 2015. Loss estimation for landslides in mountain areas- An integrated toolbox for vulnerability assessment and damage documentation. Environmental Modelling and Software, 63,: 156-169

Papathoma-Köhle M. 2016. Vulnerability curves vs vulnerability indicators: application of an indicator-based methodology for debris-flow hazards. NHESS, 16(8): 1771-1790.

Thouret et al., 2014. Assessing physical vulnerability in large cities exposed to flash floods and debris flow: the case of Arequipa (Peru). Natural Hazards, 73: 1771-1815

Leelawat, N. Suppasri A., Charvet I., Imamura F., 2014. Building damage from the 2011 Great East Japan tsunami: quantitative assessment of influential factors. Natural hazards, 73: 449-471.

But also similar studies looking at the connection of social variables to the consequences of natural hazards:

Adger N., 1998. Indicators of social and economic vulnerability to climate change in Vietnam. CSERGE Working Paper GEC 98-02

Cutter S. 2003. Social vulnerability to environmental hazards

Adger et al., 2004. New indicators of vulnerability and adaptive capacity. Tyndall Project IT 1.11: July 2001-June 2003. Final project report.

Connection to these works is essential for two reasons: first the existing literature review gap will be filled and second the aim of the study will be better understood since the results of the study may have a direct practical application.

3. The authors refer to the implementation of the European flood directive in Germany. This is an interesting point which remains which may be connected to the first comment above: Flash floods and surface water flooding are according to the authors neglected by the directive. How can the presented research fill this gap? Debris flow is actually a landslide type so naturally is not covered by the flood directive. Moreover, it has been often pointed out that during an event more than one processes may affect the elements at risk. See and refer to Totschnig et al (2011) who claim that: "During one individual event, the respective processes in the torrent often change due to the temporal and spatial variability of sediment concentration".

Totschnig R., Sedlacek W., Fuchs S., 2011. A quantitative vulnerability function for fluvial sediment transport. Natural Hazards, 58: 681-703.

4. According to the authors, the intensity of the process derives from the following two factors: -The inundation depth -The exposition of the building in flow direction. In my opinion the second factor is not relevant to the process and should not be considered in the process intensity (the intensity of the rain is the same either a person holds an umbrella or not, right?). Moreover, the intensity of the "flash flood" or "debris flow" also depends on other factors such as the velocity, the viscosity and the material that the flow contains, their size and percentage in the water.

Specific comments:

Title: The title is too general and does not reflect the content of the paper.

Abstract: why is the understanding of damage so important? What can you do with the expected results? Who may use them and how?

Introduction: Page 2, last paragraph. The authors claim that the aim of this brief communication is twofold, however, they present three aims in the following paragraphs: 1) identifying factors that govern damage, 2) the methods and the analysis of the factors 3) advantages of open source software. Additionally, the practical application of the results should also be included here. The aim of the brief communication is not clear and my feeling is that the authors do or actually present too much for a brief communication but not enough for a full research paper.

Methods: -In the first paragraph there is a reference to debris flow. Please, clarify what is the process that is investigated here. -Structural precaution: how can we check the correlation here? Shouldn't each precaution measure be a variable itself with YES/NO? -Higher ground level: Is the higher ground level always related to low damages? What about the effects of erosion during such an event? In a paper (partly by the same authors) describing the event (Agarwal et al 2016. Die Sturzflut in Braunsbach, Mai 2016. Eine Bestandsaufnahme und Ereignisbeschreibung) we can see pictures (page 3, figure 1, central photo) showing houses that have been damaged not because of a high debris level but because of erosion. In this case even if the ground floor is elevated the damage is still significant. How do you address this issues here? This also connects to a previous comment. I believe that the choice of variables has to be explained and discussed at the beginning.

Results: -Page 3, line 30: how is this database unbiased? A large proportion of the characteristics of each variable depends on expert judgement. Why does this have a minor impact (p.4, line 2)? -page 4, line 22: delete repeated word ("this"). - Why so many methods for the correlation tests? Why these specific ones and not another

[Figure]

one (e.g. Mann Whitney U test)? (explain in the "Methods" chapter) -In subchapter 3.1 reference to Figure 1 is needed. -p.5, line 3: the authors refer to the intensity of the process which is characterized by the "inundation depth" and the "exposition of building in flow direction". What about other characteristics such as flow velocity or sediment content? Aren't these characteristics related to the impact on buildings? The exposition of the building in flow direction has to do with the orientation of the building itself and not with the process...Is it correct to consider it as a defining factor for the intensity?

Table 1: why do you include categories with no representative buildings? (e.g. Rubber or steel buildings and terraced houses, conservatory, greenhouse, chemical and sewage contamination). Is the list of variables exhaustive? Figure 1: Are all the variables for table 1 included in the correlation test? If not, why not? What is with the "cellar"? The "estimated construction year"? In page 3, line 27 you refer to 21 variables, yet in Figure 1 there are only 14. Figure 2: it is not clear how the process intensity map has derived. Is the inundation depth and the exposition of building in flow direction equally important in defining the intensity? Is the intensity lower where there is no building to be exposed to the flow direction? How can this map use and what do we learn out of this map? Please refer to the following:

Fuchs S., Ornetsmüller C., Totschnig R. 2012. Spatial scan statistics in vulnerability assessment: an application to mountain hazards. Natural Hazards, 64: 2129-2151.

Fuchs et al. (2012) also detected spatial distribution patterns of loss ratios in four torrent fans in Austria.

Please also note the supplement to this comment:
http://www.nat-hazards-earth-syst-sci-discuss.net/nhess-2016-387/nhess-2016-387-RC1-supplement.pdf

---

## Referee Comment (RC2) · Anonymous Referee #2 · 23 Jan 2017

The paper describes the field survey and some first results after the flash flood event of Braunsbach in Baden-Württemberg in Germany. This type of event and the analyses are an interesting and relevant topic in the field of building damage due to extreme flood events. The complex characteristic of these extreme flood events and the resulting, in some cases, very heavy structural damage is not only in Germany an insufficient understood problem.

The aims of the paper are the identification of the damage relevant parameter due to flash floods and a discussion about the benefits of the use of the open source software "KoBoCollect" for the data acquisition.

The paper gives a short overview about the process of the event and the investigation

area. The relevant aspects of preparation and realization of the data collection during the field survey are described.

During the field survey, the authors classified the damaged buildings into a damage classification system developed by other authors. A damage grade as a measure for the structural damage was assigned to each damage case. These damage grades and the documented impact and building parameter are the basis for the statistical analyses for the identification of the damage-relevant parameters. These statistical analyses are a further focus of the paper. From the viewpoint of the referee, the linkages between the individual steps of the described procedure are logical and comprehensible.

The principle problem of the paper is mentioned by the first referee. A "brief communication" should represent a significant contribution to science, ground breaking and new results . . .

In its present form the paper would be in principle a good damage report after correcting some inaccuracies. But in the present form it fulfils not the demand for a brief communication. In general there are two possibilities: to find a journal that accepts a report form or like suggested by the first referee, to extend the work to a research paper including a detailed analysis with more graphs and figures. In the latter case also more topic related literature should be cited. In each case the type of impact (flash flood, debris flow or mud flow) should be clearly sepa-rated with respect to the involved material components.

Some other comments are necessary: By the application of the damage classification system, the authors speak from the assignment of damage classes or degree of damage. In contrast, the original publication refers to the term "damage grades".

According the paper, the team was in the field first one week after the event. This is related with the careful preparations before the survey. However, it should be discussed whether the damages a week after the event still clearly assessable due to the advanced clean-up work. It could be also discussed, whether the water level measurements with the thermographic camera the ascending humidity in the walls was taken into account.

A discussion about the topic process intensity seems also necessary. The first referee has here the opinion that the exposition belongs not to intensity. I believe at the end this is a question of the understanding of the meaning of intensity. Should the intensity considered only as a combination of impact parameter (water level, velocity, material density and debris impact)? Or has it an extended meaning like for earthquake according to EMS-98 (Grünthal et al. 1998), where also the effects on humans, nature and building were considered for the assignment of the intensity? Clear, for the damage also the exposition of the building can be relevant (Maiwald & Schwarz, 2015). A high exposition leads by such dynamic impact characteristics to higher loads on the buildings. With respect to these dynamic impacts especially the legitimation of the replacement of mean water level for some calculated percentiles with the exposition classes is unclear. Is there really a meaningful correlation?

It could be not expected, that these complex topic can be analysed in a really detailed form from a limited study of 96 damage cases. Therefore is more comprehensive data base necessary. But after a major revision of this paper and its extension to a research paper we can expect more detailed insights in the topic. I look forward to the further progress of the work.

Best regards

Literature:

Grünthal, G. (ed.), Musson, R., Schwarz, J., Stucchi, M. (1998): European Macroseismic Scale 1998. Cahiers de Centre Européen de Géodynamique et de Séismologie, Volume 15, Luxembourg)

Maiwald, H., Schwarz, J. (2015): Damage And Loss Prognosis Tools Correlating Flood Action And Building's Resistance-type Parameters, International Journal of Safety and

Security Engineering, Volume 5 (2015), Issue 3, 222 - 250

---

## Author Comment (AC1) · 8 Mar 2017

Author: Jonas Laudan 1

Co-authors: Viktor Rözer 2, Tobias Sieg 1/2, Kristin Vogel 1, Annegret H. Thieken 1

1 University of Potsdam, Institute of Earth and Environmental Science, Karl-Liebknecht-Strasse 24-25, 14476 Potsdam, Germany

2 GFZ German Research Centre for Geosciences, Department of Hydrology, Telegrafenberg, 14473 Potsdam, Germany
* * *
General answer: We thank the reviewer for the helpful and constructive comments as well as the reasonable overall suggestion to transform the Brief Communication into a Research paper. We acknowledge the reviewer's suggestions, which in many cases could not be implemented in the submitted version due to the chosen paper format. That especially holds for a more comprehensive literature review and an embedding of existing studies as well as a detailed outline of our methods and results. Thus, we would like to follow the reviewer's suggestion to transform the Brief Communication into a Research paper, addressing all helpful comments.

General comments of the reviewer

Reviewer quote 1: It is not very clear for which process the research has been conducted for. The title refers to "flash floods", however, in the text the term "debris flow" is often used for the process under investigation (e.g. p. 2, line 31). Are these two processes identical for the authors? What is the difference of these processes regarding their impact on buildings? Were all the buildings under investigation impacted by the same process?

Answer 1: The presented research aims to identify damaging processes related to flash floods, which can trigger debris flows to a certain degree. The flash flood in Braunsbach, was accompanied by a considerable amount of sediment, boulders and rubble, potentially showing flow characteristics of debris flows as defined by Fuchs et al. (2010) and Borga et al. (2014). Yet, a clear distinction between flash floods and debris flows is not always straightforward. In the revised version of the paper, we will clearly define flash floods and debris flows and we aim for consistency and adequate wording for the process. All buildings under investigation were affected by the same primary process, namely flash flood. However, the damage patterns are highly influenced by the amount and force of transported debris colliding with building walls and damaging the building structure.

Reviewer quote 2: The article should refer to similar studies and their connection to

them, for example: Papathoma-Köhle M., Zischg A., Fuchs, S. Glade T., Keiler M. 2015. Loss estimation for landslides in mountain areas- An integrated toolbox for vulnerability assessment and damage documentation. Environmental Modelling and Software, 63,: 156-169 Papathoma-Köhle M. 2016. Vulnerability curves vs vulnerability indicators: application of an indicator-based methodology for debris-flow hazards. NHESS, 16(8): 1771-1790. Thouret et al., 2014. Assessing physical vulnerability in large cities exposed to flash floods and debris flow: the case of Arequipa (Peru). Natural Hazards, 73: 1771-1815 Leelawat, N. Suppasri A., Charvet I., Imamura F., 2014. Building damage from the 2011 Great East Japan tsunami: quantitative assessment of influential factors. Natural hazards, 73: 449-471. But also similar studies looking at the connection of social variables to the consequences of natural hazards: Adger N., 1998. Indicators of social and economic vulnerability to climate change in Vietnam. CSERGE Working Paper GEC 98-02 Cutter S. 2003. Social vulnerability to environmental hazards Adger et al., 2004. New indicators of vulnerability and adaptive capacity. Tyndall Project IT 1.11: July 2001-June 2003. Final project report. Connection to these works is essential for two reasons: first the existing literature review gap will be filled and second the aim of the study will be better understood since the results of the study may have a direct practical application.

Answer 2: Thank you for your suggestions. As stated above, when converting the brief communication to a research paper, we will take more existing literature into account to integrate our work into an up-to-date, scientific framework. This will include a review section

Reviewer quote 3: The authors refer to the implementation of the European flood directive in Germany. This is an interesting point which remains which may be connected to the first comment above: Flash floods and surface water flooding are according to the authors neglected by the directive. How can the presented research fill this gap? Debris flow is actually a landslide type so naturally is not covered by the flood directive. Moreover, it has been often pointed out that during an event more than one processes

may affect the elements at risk. See and refer to Totschnig et al (2011) who claim that: "During one individual event, the respective processes in the torrent often change due to the temporal and spatial variability of sediment concentration". Totschnig R., Sedlacek W., Fuchs S., 2011. A quantitative vulnerability function for fluvial sediment transport. Natural Hazards, 58: 681-703.

Answer 3: Thank you for your suggestions. The implementation of the European Floods Directive 2007/60/ EC in Germany and the implications according to the German Federal Water Act (e.g. the obligation for flood adapted spatial planning and the creation of flood risk maps) will be discussed in the revised paper. The consideration of flash floods as a "significant risk" would have serious implications on mapping, planning and risk management. This holds especially for the not yet mandatory creation of flood hazard risk and risk maps, which, in case of flash floods, do currently not exist nationwide and would have to be generated. As a further consequence, German Federal Water Act intends a building ban in all areas that are affected by a 100-year flood event. Therefore, the consideration of flash floods or surface water flooding could have serious consequences for local planning. As already mentioned, the particular flash flood in Braunsbach revealed the complexity and the high impact of such events. Even if flash floods are technically not considered as significant hazards in the German Federal Water Act, it is possible to include measures for reducing their impacts (e.g. potential tangible as well as intangible damage) in flood risk management plans. The current state of the still ongoing discussion will be addressed in the revised paper.

Reviewer quote 4: According to the authors, the intensity of the process derives from the following two factors: -The inundation depth -The exposition of the building in flow direction. In my opinion the second factor is not relevant to the process and should not be considered in the process intensity (the intensity of the rain is the same either a person holds an umbrella or not, right?). Moreover, the intensity of the "flash flood" or "debris flow" also depends on other factors such as the velocity, the viscosity and the material that the flow contains, their size and percentage in the water.

Answer 4: With process intensity we refer to potential (external) factors affecting the building and resulting in damage, which are independent from the building characteristics and can be surveyed in the aftermath of the event. Since these factors (i.e. the inundation depth as well as the exposition) relate to the buildings relative position, a combined value - "process intensity" - should indicate the potential physical impact and intensity of the flash flood at the particular house or in a specified area for that matter (i.e. process intensity map). With this term we do not refer to the physical characteristics of the flash flood itself or flood inherent processes (flow velocities, duration), since these processes could not be determined on site. Instead, we give a proxy for the flow impacts and the impact forces of flow and debris on a building. However, to differentiate from flood inherent processes, we will use the term "local impact" in the revised paper.

Specific comments of the reviewer

Reviewer quote on the title and abstract: Title: The title is too general and does not reflect the content of the paper. Abstract: why is the understanding of damage so important? What can you do with the expected results? Who may use them and how?

Answer on the title and abstract: Title: Thank you for the hint. The paper title will be revised that it reflects the content of the paper in a more elaborate way. One option could be: "Damage assessment after the flash flood in Braunsbach 2016: A data collection and analysis for an improved understanding of damaging processes during flash floods." Abstract: The understanding of damage caused by flash floods is of great interest because we observe changing weather patterns in Central Europe and Germany due to the climate change (see Murawski, A., Zimmer, J. and Merz, B. 2016: High spatial and temporal organization of changes in precipitation over Germany for 1951-2006. International Journal of Climatology, 36, 6, 2582-2597. doi:10.1002/joc.4514. Volosciuk, C., Maraun, D., Semenov, VA., Tilinina, N., Gulev, SK. and Latif, M. 2016: Rising Mediterranean Sea Surface Temperatures Amplify Extreme Summer Precipitation in Central Europe. Scientific Reports, 6, 32450. doi:10.1038/srep32450. Beniston,

M., Stephenson, DB., Christensen, OB., Ferro, CAT., Frei, C., Goyette, S., Halsnaes, K., Holt, T., Jylha, K., Koffi, B., Palutikof, J., Scholl, R., Semmler, T. and Woth, K. 2007: Future extreme events in European climate: an exploration of regional climate model projections. Climatic Change, 81, 71-95. doi: 10.1007/s10584-006-9226-z.). As an effect of higher precipitation intensities within shorter time periods, flash floods, such as seen in Braunsbach and the surrounding villages, might occur more frequently, which raises questions regarding damage mitigation, insurance and risk management in flash flood prone regions. We hope that our results contribute to a better understanding of damage driving factors with regard to those extreme events and thus enable the implementation of adequate risk reduction measures.

Reviewer quote on the introduction: Introduction: Page 2, last paragraph. The authors claim that the aim of this brief communication is twofold, however, they present three aims in the following paragraphs: 1) identifying factors that govern damage, 2) the methods and the analysis of the factors 3) advantages of open source software. Additionally, the practical application of the results should also be included here. The aim of the brief communication is not clear and my feeling is that the authors do or actually present too much for a brief communication but not enough for a full research paper.

Answer on the introduction: The aims of the brief communication were stated to be twofold, since we present two major topics. 1. The identification of factors and processes that govern the damage caused by this particular flash flood to improve the general process understanding. 2. The use of open source software, how it was implemented, carried out on site and presenting advantages as well as disadvantages. We did not declare the methods and also subsequent analysis and processing of identified factors (i.e. the process intensity/local impact map) as an aim on its own because they eventually lead to a better understanding of damage processes, what is seen as the actual aim. In the following, we present an option to revise the last introduction paragraph and express our aims in a clearer way: "This research paper follows two major

objectives. Using the flash flood in Braunsbach as a case study, it is aimed at identifying, analysing and discussing factors that govern damage caused by flash floods. As a second issue, the methods used for the ex-post damage data collection in Braunsbach and the creation of this database are presented and discussed. Since "KoBoCollect" turned out to provide major advantages with regard to the duration of data acquisition, simplicity, effectiveness and in-field handling, we demonstrate the benefits as well as important issues of open source software associated with its use. Within this regard we aim for an increased awareness of open source software and its potential for scientific and public data collection."

Reviewer quote on the methods: Methods: -In the first paragraph there is a reference to debris flow. Please, clarify what is the process that is investigated here. -Structural precaution: how can we check the correlation here? Shouldn't each precaution measure be a variable itself with YES/NO? -Higher ground level: Is the higher ground level always related to low damages? What about the effects of erosion during such an event? In a paper (partly by the same authors) describing the event (Agarwal et al 2016. Die Sturzflut in Braunsbach, Mai 2016. Eine Bestandsaufnahme und Ereignisbeschreibung) we can see pictures (page 3, figure 1, central photo) showing houses that have been damaged not because of a high debris level but because of erosion. In this case even if the ground floor is elevated the damage is still significant. How do you address this issues here? This also connects to a previous comment. I believe that the choice of variables has to be explained and discussed at the beginning.

Answer on the methods: As mentioned above, we will be consistent in the use of terms in the revised paper. The flash flood of Braunsbach, as well as several other flash floods that occurred in spring 2016, were accompanied by high erosion rates leading to high sediment and debris loads in the surface runoff. See Answer 1 at "General comments of the reviewer". Indeed, the variables for each structural precaution measure exist in a binary format, allowing for basic correlation tests. The Spearman's rho correlation of -0.11 (p 0.33) between higher ground floor and the damage class indicates

a minor damage reducing effect of higher ground levels. Yet, this effect is assumed to predominantly hold for water levels below a certain threshold (e.g. the height of the elevated ground floor) and potentially avoids the infiltration of water and sediments in these cases. As can be seen in the figure the reviewer refers to, processes such as flow velocities or erosion contribute to the building damage. However, since variables as the flow velocity and amount of transported material could not reliably be observed in the aftermath of the event, the exposition of the building was used as a proxy instead. Low exposition is often related to reduced flow velocities and to a lesser degree of sediment/debris load, which in turn leads to smaller erosion rates and less collision damage. The overall damage pattern implies that higher damage is mainly governed by higher expositions in flow direction (and probably higher flow velocities) and higher water levels. In the revised paper, we will analyse and discuss all recorded variables (see "Answer on tables and figures"). Further, we will state our motivation and variable choice for particular tests in the beginning.

Reviewer quote on the results: Results: -Page 3, line 30: how is this database unbiased? A large proportion of the characteristics of each variable depends on expert judgement. Why does this have a minor impact (p.4, line 2)? -page 4, line 22: delete repeated word ("this"). – Why so many methods for the correlation tests? Why these specific ones and not another one (e.g. Mann Whitney U test)? (explain in the "Methods" chapter) -In subchapter 3.1 reference to Figure 1 is needed. -p.5, line 3: the authors refer to the intensity of the process which is characterized by the "inundation depth" and the "exposition of building in flow direction". What about other characteristics such as flow velocity or sediment content? Aren't these characteristics related to the impact on buildings? The exposition of the building in flow direction has to do with the orientation of the building itself and not with the process: : :Is it correct to consider it as a defining factor for the intensity?

Answer on the results: With "unbiased" we refer to variations in the dataset caused by intersubjective differences in classification. An alternative description such as "consistency among different team members during the data collection" might be better to describe this issue. We consider the data to be consistent in a way that the team members had very similar opinions e.g. on the damage classes or exposition in flow direction. Thus, a bias in the dataset due to personal variations in expert judgement is expected to be low. We chose the Spearman's rho correlation test, since we are interested in a correlation measurement of variables with different measurement scales and distributions. The Mann Whitney U test is primarily used to compare two datasets from the same population which does not seem to be beneficial for our purposes to describe variable coherences. As described above, flow velocity and sediment load could not be determined on site. Instead, exposition of the building was used to respect both parameters. Yet, the used term "process intensity" will be revised. We will rather use the term "local impact" to differentiate from flood inherent processes. However, as a difference to riverine flooding, we revealed that, during this flash flood, the physical impact (caused by debris, boulders and/or rubble) on buildings holds great importance as a damage driving factor and is dependent on both, the buildings relative position to the stream and probably shielding effects of neighbouring buildings. In general it can be said that, with "process intensity"/"local impact", we do not focus on the hazard itself but rather analyse the consequences of flash floods and circumstances which are related to higher damage.

Reviewer quote on tables and figures: Table 1: why do you include categories with no representative buildings? (e.g. Rubber or steel buildings and terraced houses, conservatory, greenhouse, chemical and sewage contamination). Is the list of variables exhaustive? Figure 1: Are all the variables for table 1 included in the correlation test? If not, why not? What is with the "cellar"? The "estimated construction year"? In page 3, line 27 you refer to 21 variables, yet in Figure 1 there are only 14. Figure 2: it is not clear how the process intensity map has derived. Is the inundation depth and the exposition of building in flow direction equally important in defining the intensity? Is the intensity lower where there is no building to be exposed to the flow direction? How can this map use and what do we learn out of this map? Please refer to the following:

Fuchs S., Ornetsmüller C., Totschnig R. 2012. Spatial scan statistics in vulnerability assessment: an application to mountain hazards. Natural Hazards, 64: 2129-2151. Fuchs et al. (2012) also detected spatial distribution patterns of loss ratios in four torrent fans in Austria.

Answer on tables and figures: To avoid offline and unsynchronised modifications, several variable categories were included in the questionnaire beforehand, without knowledge about the specific situation in the research area. We consider the presentation of the complete set of possible answers to be relevant for comparisons with followup studies. The table represents the complete survey, as it was designed and the distribution of occurrences, categories with zero cases are thus included as well. Further, the complete survey contains 22 variables, not 21. These mistakes will be corrected. Due to the paper format, not all variables were included in the correlation tests, only those which seem most rewarding with regard to coherences and the desired motivation. Our main objectives were to analyse the damage driving factors of flash floods and to reveal potential differences compared to riverine flooding. E.g. Maiwald et al. 2015 give an overview of known factors which influence structural damage on buildings. Especially the building material, condition (before the event) and the age are important factors related to the buildings resistance potential. Factors such as inundation level and contamination relate to "action" parameters (Maiwald et al. 2015) and describe external forces. Thus, the choice of the analysed variables was based on both, existing literature as well as expert judgement, i.e. including the exposition in flow direction, the abundance of large shop windows on the ground level or the sealing of the near environment as well. However, to use the full potential of our database, we will analyse all recorded variables in the revised paper. The "process intensity"/"local impact", which is a combination of the inundation depth measured at the building and the building's exposition, can be seen as a proxy for local flood related impact forces. Since both variables show the same correlation value to the caused damage and are further rated to be equally important in both developed damage models (RGLM and RF), we chose a combination of these factors, where both contribute to equal extents. While the inundation depth has continuous values, which are roughly uniformly distributed between 2 and 360 cm, the exposition in flow direction is recorded in three classes (low, medium, high). To achieve comparable variable ranges, the exposition classes "low", "medium" and "high" are transformed into the mean values of the lower, middle and upper third of recorded water levels. The derived values 57, 133 and 230 fit into the range of observed water levels, enabling a combination of both attributes. The calculated "process intensity"/"local impact" corresponds to the sum of water level and transformed exposition value. Please note that the exposition values are not used to replace water levels, but are only transformed into a comparable range. In the revised paper we will illustrate our methods with graphics for a better understanding. The "process intensity"/"local impact"-map was created in QGIS (Figure 2) and fills the gaps between buildings with interpolated values to express a hypothetical intensity for hypothetical buildings at that spot. In the framework of the paper, the map is used to illustrate the flash flood process in Braunsbach and to underline the impact of water depth and exposition on the resulting damage. Overall the estimation of a local impact could be used in strategic planning of mitigation measures against future hazards in Braunsbach. The same approach can also be used for similar villages in that region, given that information about the potential flash flood (e.g. inundation depth) is available either from observations of an actual event or from flash flood models. The consideration of exposition as damage driver fits to the statement by Fuchs et al. (2012), saying that the general land use and settlement patterns play an important role in the geographical distribution of building damage. Thus, our map may contribute to the identification of potentially vulnerable locations on a small scale and within case studies.

---

## Author Comment (AC2) · 8 Mar 2017

Author: Jonas Laudan 1

Co-authors: Viktor Rözer 2, Tobias Sieg 1/2, Kristin Vogel 1, Annegret H. Thieken 1

1 University of Potsdam, Institute of Earth and Environmental Science, Karl-Liebknecht-Strasse 24-25, 14476 Potsdam, Germany

2 GFZ German Research Centre for Geosciences, Department of Hydrology, Telegrafenberg, 14473 Potsdam, Germany
* * *
General answer: We thank the reviewer for the constructive comments. As stated by referee 1 as well, we agree with the reviewer that the Brief Communication should be transformed into a Research paper, in which we extend our analysis and discussion.

Comments of the reviewer

Reviewer quote, paragraph 1, 2, 3 & 4: The paper describes the field survey and some first results after the flash flood event of Braunsbach in Baden-Württemberg in Germany. This type of event and the analyses are an interesting and relevant topic in the field of building damage due to extreme flood events. The complex characteristic of these extreme flood events and the resulting, in some cases, very heavy structural damage is not only in Germany an insufficient understood problem. The aims of the paper are the identification of the damage relevant parameter due to flash floods and a discussion about the benefits of the use of the open source software "KoBoCollect" for the data acquisition. The paper gives a short overview about the process of the event and the investigation area. The relevant aspects of preparation and realization of the data collection during the field survey are described. During the field survey, the authors classified the damaged buildings into a damage classification system developed by other authors. A damage grade as a measure for the structural damage was assigned to each damage case. These damage grades and the documented impact and building parameter are the basis for the statistical analyses for the identification of the damage-relevant parameters. These statistical analyses are a further focus of the paper. From the viewpoint of the referee, the linkages between the individual steps of the described procedure are logical and comprehensible.

Answer 1, 2, 3 & 4: Thank you for acknowledging the relevance of our work. In our research we used the damage classification scheme developed by Schwarz and Maiwald (2007) in order to ensure comparability to other studies. In a revised version, we will keep the general outline and extend our research as well as literature review.

Reviewer quote, paragraph 5 & 6: The principle problem of the paper is mentioned

by the first referee. A "brief communication" should represent a significant contribution to science, ground breaking and new results... In its present form the paper would be in principle a good damage report after correcting some inaccuracies. But in the present form it fulfils not the demand for a brief communication. In general there are two possibilities: to find a journal that accepts a report form or like suggested by the first referee, to extend the work to a research paper including a detailed analysis with more graphs and figures. In the latter case also more topic related literature should be cited. In each case the type of impact (flash flood, debris flow or mud flow) should be clearly separated with respect to the involved material components.

Answer 5 & 6: Thank you for this suggestion. As also stated in our response to the comments of reviewer 1, we agree that the conversion of the Brief Communication into a Research paper is a helpful suggestion. We will include more graphics supporting our methods and analysis. These will be 1. Additional maps which display certain attributes of buildings and their surroundings ( i.e. building usage, near surrounding sealed and/or inundation depth) 2. Graphics to explain our methods (i.e. the derivation of the process intensity/local impact in the revised paper) 3. Graphics related to more detailed analysis Our detailed analysis will include correlation tests with all recorded variables. We consider additional analysis (i.e. multinomial logistic regression) to obtain additional outcomes and deepen the discussion. Further, we will discuss our work in the context of existing studies on this topic. The flash flood in Braunsbach was accompanied by a considerable amount of sediment and building rubble, potentially showing flow characteristics of debris flows such as defined by Fuchs et al. 2010 and Borga et al 2014. Yet, a clear distinction between flash floods and debris flows is not always straightforward. In the revised version of the paper, we will clearly define flash floods and debris flows and we aim for consistency and adequate wording for the process.

Reviewer quote, paragraph 7: Some other comments are necessary: By the application of the damage classification system, the authors speak from the assignment of damage

classes or degree of damage. In contrast, the original publication refers to the term "damage grades".

Answer 7: Thank you for the comment, we will be more careful and consistent with the terms we use to describe the damage classification. In a revised version, we will use the term "damage grades".

Reviewer quote, paragraph 8: According the paper, the team was in the field first one week after the event. This is related with the careful preparations before the survey. However, it should be discussed whether the damages a week after the event still clearly assessable due to the advanced clean-up work. It could be also discussed, whether the water level measurements with the thermographic camera the ascending humidity in the walls was taken into account.

Answer 8: One week after the event, the structural damage on buildings and building characteristics was still assessable, since the main work within this period was mainly focused on clearing the roads, establishing paths for large construction machinery as well as removing and cleaning the interior of affected buildings. Some areas of the village were even not accessible before, since roads were blocked by debris and buildings in danger of collapse had to be secured. The progress of the clean-up work was further beneficial for the damage assessment, as a thick layer of debris and rubble previously covered big parts of the building damage. The use of the thermographic camera will be better presented and discussed in the revised paper, since it offered advantages in such cases, where the inundation depth could not be reliably estimated. We agree with the reviewer, that the ascending humidity in the walls is a point to consider when using a thermographic camera for water level estimations. For that reason the thermal images were mainly used to verify estimations based on visible mud contamination and marks caused by water and transported debris. Since the thermally derived water levels matched well with visible traces, thermal images were also used to estimate the inundation depth for buildings with no or little visible traces.

Reviewer quote, paragraph 9: A discussion about the topic process intensity seems also necessary. The first referee has here the opinion that the exposition belongs not to intensity. I believe at the end this is a question of the understanding of the meaning of intensity. Should the intensity considered only as a combination of impact parameter (water level, velocity, material density and debris impact)? Or has it an extended meaning like for earthquake according to EMS-98 (Grünthal et al. 1998), where also the effects on humans, nature and building were considered for the assignment of the intensity? Clear, for the damage also the exposition of the building can be relevant (Maiwald & Schwarz, 2015). A high exposition leads by such dynamic impact characteristics to higher loads on the buildings. With respect to these dynamic impacts especially the legitimation of the replacement of mean water level for some calculated percentiles with the exposition classes is unclear. Is there really a meaningful correlation?

Answer 9: We agree with the statement of the reviewer that the attributes of intensity are determined by different meanings and can include exposition grades as well. In our case, the process intensity includes factors which are independent from building characteristics and do not only represent flood inherent parameters. As stated in the answer on the first referee comment as well, we will replace the term "process intensity" with "local intensity", to reflect the actual meaning in a better way. The exposition classes needed to be transformed in order to derive the process intensity. Still, the inundation depth and the exposition classes contribute independently to the process intensity. In the following paragraph, the methods are described in detail (copy from the answers on referee comment RC1): The "process intensity"/"local impact", which is a combination of the inundation depth measured at the building and the building's exposition, can be seen as a proxy for local flood related impact forces. Since both variables show the same correlation value to the caused damage and are further rated to be equally important in both developed damage models (RGLM and RF), we chose a combination of these factors, where both contribute to equal extents. While the inundation depth has continuous values, which are roughly uniformly distributed between 2

and 360 cm, the exposition in flow direction is recorded in three classes (low, medium, high). To achieve comparable variable ranges, the exposition classes "low", "medium" and "high" are transformed into the mean values of the lower, middle and upper third of recorded water levels. The derived values 57, 133 and 230 fit into the range of observed water levels, enabling a combination of both attributes. The calculated "process intensity"/"local impact" corresponds to the sum of water level and transformed exposition value. Please note that the exposition values are not used to replace water levels, but are only transformed into a comparable range. In the revised paper we will illustrate our methods with graphics for a better understanding.

Reviewer quote, paragraph 10: It could be not expected, that these complex topic can be analysed in a really detailed form from a limited study of 96 damage cases. Therefore is more comprehensive data base necessary. But after a major revision of this paper and its extension to a research paper we can expect more detailed insights in the topic. I look forward to the further progress of the work.

Answer 10: We thank the reviewer and aim to contribute to a better understanding of flash floods and related damage processes in general. Considering the fact that our research in Braunsbach resembles a case study, the 96 damage cases represent a complete inquiry. Thus, we do not claim to perform an extensive study of this complex topic. However, we will add to the knowledge within this field of research. Especially since further research is planned and an additional survey related to flash floods will be carried out soon, connecting to our results.

---

## Author Response (AR1)

**Final Author comments to:**
**Anonymous Referee #1 RC1: nhess-2016-387-RC1, 2017**
**Anonymous Referee #2 RC2: nhess-2016-387-RC2, 2017**

Author: Jonas Laudan[1]
    Co-authors: Viktor Rözer[2], Tobias Sieg[1/2], Kristin Vogel[1], Annegret H. Thieken[1]

[1]University of Potsdam, Institute of Earth and Environmental Science, Karl-Liebknecht-Strasse 24-25, 14476 Potsdam, Germany
[2]GFZ German Research Centre for Geosciences, Department of Hydrology, Telegrafenberg, 14473 Potsdam, Germany

**Referee #1**

**General answer:**

We thank the reviewer for the helpful and constructive comments as well as the reasonable overall suggestion to transform the Brief Communication into a Research paper. We acknowledge the reviewer's suggestions, which in many cases could not be implemented in the submitted version due to the chosen paper format. That especially holds for a more comprehensive literature review and an embedding of existing studies as well as a detailed outline of our methods and results. Thus, we followed the reviewer's suggestion to transform the Brief Communication into a Research paper, addressing all helpful comments.

**General comments of the reviewer 1**

**Reviewer1 quote 1:**

It is not very clear for which process the research has been conducted for. The title refers to "flash floods", however, in the text the term "debris flow" is often used for the process under investigation (e.g. p. 2, line 31). Are these two processes identical for the authors? What is the difference of these processes regarding their impact on buildings? Were all the buildings under investigation impacted by the same process?

**Answer 1:**

The presented research aims to identify damaging processes related to flash floods, which can trigger debris flows to a certain degree. The flash flood in Braunsbach, was accompanied by a considerable amount of sediment, boulders and rubble, potentially showing flow characteristics of debris flows as defined by Fuchs et al. (2010) and Borga et al. (2014). Yet, a clear distinction between flash floods and debris flows is not always straightforward. In the revised version of the paper, we clearly defined flash floods and debris flows and we aimed for consistency and adequate wording for the process.

All buildings under investigation were affected by the same primary process, namely flash flood. However, the damage patterns are highly influenced by the amount and force of transported debris colliding with building walls and damaging the building structure.

**Reviewer1 quote 2:**

The article should refer to similar studies and their connection to them, for example:

Papathoma-Köhle M., Zischg A., Fuchs, S. Glade T., Keiler M. 2015. Loss estimation for landslides in mountain areas- An integrated toolbox for vulnerability assessment and damage documentation. Environmental Modelling and Software, 63,: 156-169

Papathoma-Köhle M. 2016. Vulnerability curves vs vulnerability indicators: application of an indicator-based methodology for debris-flow hazards. NHESS, 16(8): 1771-1790.

Thouret et al., 2014. Assessing physical vulnerability in large cities exposed to flash floods and debris flow: the case of Arequipa (Peru). Natural Hazards, 73: 1771-1815 Leelawat, N. Suppasri A., Charvet I., Imamura F., 2014. Building damage from the 2011 Great East Japan tsunami: quantitative assessment of influential factors. Natural hazards, 73: 449-471.

But also similar studies looking at the connection of social variables to the consequences of natural hazards:

Adger N., 1998. Indicators of social and economic vulnerability to climate change in Vietnam. CSERGE Working Paper GEC 98-02

Cutter S. 2003. Social vulnerability to environmental hazards

Adger et al., 2004. New indicators of vulnerability and adaptive capacity. Tyndall Project IT 1.11: July 2001-June 2003.

Final project report.

Connection to these works is essential for two reasons: first the existing literature review gap will be filled and second the aim of the study will be better understood since the results of the study may have a direct practical application.

**Answer 2:**

Thank you for your suggestions. As stated above, while converting the brief communication to a research paper, we took more existing literature into account to integrate our work into an up-to-date, scientific framework. A review section is included in the introduction.

**Reviewer1 quote 3:**

The authors refer to the implementation of the European flood directive in Germany. This is an interesting point which
remains which may be connected to the first comment above: Flash floods and surface water flooding are according to the
authors neglected by the directive. How can the presented research fill this gap? Debris flow is actually a landslide type so
naturally is not covered by the flood directive. Moreover, it has been often pointed out that during an event more than one
processes may affect the elements at risk. See and refer to Totschnig et al (2011) who claim that: "During one individual
event, the respective processes in the torrent often change due to the temporal and spatial variability of sediment
concentration".

Totschnig R., Sedlacek W., Fuchs S., 2011. A quantitative vulnerability function for fluvial sediment transport. Natural
Hazards, 58: 681-703.

**Answer 3:**

Thank you for your suggestions. The implementation of the European Floods Directive 2007/60/ EC in Germany and the
implications according to the German Federal Water Act (e.g. the obligation for flood adapted spatial planning and the
creation of flood risk maps) is shortly mentioned in the revised paper. The consideration of flash floods as a "significant
risk" would have serious implications on mapping, planning and risk management. This holds especially for the not yet
mandatory creation of flood hazard risk and risk maps, which, in case of flash floods, do currently not exist nationwide and
would have to be generated. As a further consequence, German Federal Water Act intends a building ban in all areas that are
affected by a 100-year flood event. Therefore, the consideration of flash floods or surface water flooding could have serious
consequences for local planning.

As already mentioned, the particular flash flood in Braunsbach revealed the complexity and the high impact of such events.
Even if flash floods are technically not considered as significant hazards in the German Federal Water Act, it is possible to
include measures for reducing their impacts (e.g. potential tangible as well as intangible damage) in flood risk management
plans.

**Reviewer1 quote 4:**

According to the authors, the intensity of the process derives from the following two factors: -The inundation depth -The
exposition of the building in flow direction. In my opinion the second factor is not relevant to the process and should not be
considered in the process intensity (the intensity of the rain is the same either a person holds an umbrella or not, right?).
Moreover, the intensity of the "flash flood" or "debris flow" also depends on other factors such as the velocity, the viscosity
and the material that the flow contains, their size and percentage in the water.

**Answer 4:**

With process intensity we refer to potential (external) factors affecting the building and resulting in damage, which are independent from the building characteristics and can be surveyed in the aftermath of the event. Since these factors (i.e. the inundation depth as well as the exposition) relate to the buildings relative position, a combined value - "process intensity" - should indicate the potential physical impact and intensity of the flash flood at the particular house or in a specified area for that matter (i.e. process intensity map). With this term we do not refer to the physical characteristics of the flash flood itself or flood inherent processes (flow velocities, duration), since these processes could not be determined on site. Instead, we give a proxy for the flow impacts and the impact forces of flow and debris on a building. However, to differentiate from flood inherent processes, we use the term "local impact" in the revised paper.

**Specific comments of the reviewer 1**

**Reviewer1 quote on the title and abstract:**
Title: The title is too general and does not reflect the content of the paper.
Abstract: why is the understanding of damage so important? What can you do with the expected results? Who may use them and how?

**Answer on the title and abstract:**
Title: Thank you for the hint. The paper title has been revised that it reflects the content of the paper in a more elaborate way. The title is now: "Damage assessment in Braunsbach 2016: A data collection and analysis for an improved understanding of damaging processes during flash floods."
Abstract: The understanding of damage caused by flash floods is of great interest because we observe changing weather patterns in Central Europe and Germany due to the climate change. Further, an increased risk for higher damage due to flooding can be detected, which is mainly influenced by urbanization, economic growth as well as changing land use patterns (see European Environment Agency: Climate change, impacts and vulnerability in Europe 2016 - An indicator-based report, 1, Luxembourg, doi:10.2800/534806, 2017.

Thieken, A. H., Cammerer, H., Dobler, C., Lammel, J., and Schöberl, F.: Estimating changes in flood risks and benefits of non-structural adaptation strategies - a case study from Tyrol, Austria, Mitig Adapt Strateg Glob Change, 21, 3, 343-376, doi:10.1007/s11027-014-9602-3, 2014.

Murawski, A., Zimmer, J. and Merz, B. 2016: High spatial and temporal organization of changes in precipitation over Germany for 1951-2006. International Journal of Climatology, 36, 6, 2582-2597. doi:10.1002/joc.4514.

Volosciuk, C., Maraun, D., Semenov, VA., Tilinina, N., Gulev, SK. and Latif, M. 2016: Rising Mediterranean Sea Surface

Temperatures Amplify Extreme Summer Precipitation in Central Europe. Scientific Reports, 6, 32450. doi:10.1038/srep32450.

Beniston, M., Stephenson, DB., Christensen, OB., Ferro, CAT., Frei, C., Goyette, S., Halsnaes, K., Holt, T., Jylha, K., Koffi, B., Palutikof, J., Scholl, R., Semmler, T. and Woth, K. 2007: Future extreme events in European climate: an exploration of regional climate model projections. Climatic Change, 81, 71-95. doi: 10.1007/s10584-006-9226-z.).

As an effect of higher precipitation intensities within shorter time periods, flash floods might occur more frequently in future. In combination with changing land use patterns and urbanisation, the damage mitigation, insurance and risk management in flash flood prone regions become increasingly important. We hope that our results contribute to a better understanding of damage driving factors with regard to those extreme events and thus enable the implementation of adequate risk reduction measures.

**Reviewer1 quote on the introduction:**

Introduction: Page 2, last paragraph. The authors claim that the aim of this brief communication is twofold, however, they present three aims in the following paragraphs: 1) identifying factors that govern damage, 2) the methods and the analysis of the factors 3) advantages of open source software. Additionally, the practical application of the results should also be included here. The aim of the brief communication is not clear and my feeling is that the authors do or actually present too much for a brief communication but not enough for a full research paper.

**Answer on the introduction:**

The aims of the brief communication were stated to be twofold, since we present two major topics.

1. The identification and discussion of factors and processes that govern the damage caused by this particular flash flood to improve the general process understanding.
2. The use of open source software, how it was implemented, carried out on site and presenting advantages as well as disadvantages.

We did not declare the methods and also subsequent analysis and processing of identified factors (i.e. the process intensity/local impact map) as an aim on its own because they eventually lead to a better understanding of damage processes, what is seen as the actual aim. In the following, we present our option of choice to revise the last introduction paragraph and express our aims in a clearer way:

"Consequently, we present Braunsbach as a case study, having collected and analysed data in order to add to the knowledge in this field. This research paper follows two major objectives. Using the flash flood in Braunsbach as a case study, it is aimed at identifying, analysing, comparing and discussing factors that govern damage caused by this event, applying different linear and non-linear methods. As a second issue, the methods used for the ex-post damage data collection in

Braunsbach and the creation of this database are presented and discussed to demonstrate accompanying challenges as well as advantages."

**Reviewer1 quote on the methods:**

Methods: -In the first paragraph there is a reference to debris flow. Please, clarify what is the process that is investigated here. -Structural precaution: how can we check the correlation here? Shouldn't each precaution measure be a variable itself with YES/NO? -Higher ground level: Is the higher ground level always related to low damages? What about the effects of erosion during such an event? In a paper (partly by the same authors) describing the event (Agarwal et al 2016. Die Sturzflut in Braunsbach, Mai 2016. Eine Bestandsaufnahme und Ereignisbeschreibung) we can see pictures (page 3, figure 1, central photo) showing houses that have been damaged not because of a high debris level but because of erosion. In this case even if the ground floor is elevated the damage is still significant. How do you address this issues here? This also connects to a previous comment. I believe that the choice of variables has to be explained and discussed at the beginning.

**Answer on the methods:**

As mentioned above, we are now consistent in the use of terms in the revised paper. The flash flood of Braunsbach, as well as several other flash floods that occurred in spring 2016, were accompanied by high erosion rates leading to high sediment and debris loads in the surface runoff. See Answer 1 at "General comments of the reviewer".

Indeed, the variables for each structural precaution measure exist in a binary format, allowing for basic correlation tests.

The Spearman's rho correlation of -0.09 (p 0.43) between higher ground floor and the damage class only indicates a minor damage reducing effect of higher ground levels. Yet, this effect is assumed to predominantly hold for water levels below a certain threshold (e.g. the height of the elevated ground floor) and potentially avoids the infiltration of water and sediments in these cases.

As can be seen in the figure the reviewer refers to, processes such as flow velocities or erosion contribute to the building damage. However, since variables as the flow velocity and amount of transported material could not reliably be observed in the aftermath of the event, the exposition of the building was used as a proxy instead. Low exposition is often related to reduced flow velocities and to a lesser degree of sediment/debris load, which in turn leads to smaller erosion rates and less collision damage. The overall damage pattern implies that higher damage is mainly governed by higher expositions in flow direction (and probably higher flow velocities) and higher water levels. In the revised paper, we analysed all recorded variables (see "Answer on tables and figures"). Further, we stated our motivation and variable choice for particular tests in the beginning.

**Reviewer1 quote on the results:**

Results: -Page 3, line 30: how is this database unbiased? A large proportion of the characteristics of each variable depends on expert judgement. Why does this have a minor impact (p.4, line 2)? -page 4, line 22: delete repeated word ("this"). – Why so many methods for the correlation tests? Why these specific ones and not another one (e.g. Mann Whitney U test)? (explain in the "Methods" chapter) -In subchapter 3.1 reference to Figure 1 is needed. -p.5, line 3: the authors refer to the intensity of the process which is characterized by the "inundation depth" and the "exposition of building in flow direction". What about other characteristics such as flow velocity or sediment content? Aren't these characteristics related to the impact on buildings? The exposition of the building in flow direction has to do with the orientation of the building itself and not with the process: : :Is it correct to consider it as a defining factor for the intensity?

**Answer on the results:**

With "unbiased" we refer to variations in the dataset caused by intersubjective differences in classification. An alternative description such as "consistency among different team members during the data collection" might be better to describe this issue. We consider the data to be consistent in a way that the team members had very similar opinions e.g. on the damage classes or exposition in flow direction. Thus, a bias in the dataset due to personal variations in expert judgement is expected to be low.

We chose the Spearman's rho correlation test, since we are interested in a correlation measurement of variables with different measurement scales and distributions. The Mann Whitney U test is primarily used to compare two datasets from the same population which does not seem to be beneficial for our purposes to describe variable coherences.

As described above, flow velocity and sediment load could not be determined on site. Instead, exposition of the building was used to respect both parameters. Yet, the used term "process intensity" has been revised. We use the term "local impact" to differentiate from flood inherent processes. However, as a difference to riverine flooding, we revealed that, during this flash flood, the physical impact (caused by debris, boulders and/or rubble) on buildings holds great importance as a damage driving factor and is dependent on both, the buildings relative position to the stream and probably shielding effects of neighbouring buildings. In general it can be said that, with "process intensity"/"local impact", we do not focus on the hazard itself but rather analyse the consequences of flash floods and circumstances which are related to higher damage.

**Reviewer1 quote on tables and figures:**

Table 1: why do you include categories with no representative buildings? (e.g. Rubber or steel buildings and terraced houses, conservatory, greenhouse, chemical and sewage contamination). Is the list of variables exhaustive? Figure 1: Are all the variables for table 1 included in the correlation test? If not, why not? What is with the "cellar"? The "estimated construction year"? In page 3, line 27 you refer to 21 variables, yet in Figure 1 there are only 14. Figure 2: it is not clear how the process intensity map has derived. Is the inundation depth and the exposition of building in flow direction equally important in
defining the intensity? Is the intensity lower where there is no building to be exposed to the flow direction? How can this
map use and what do we learn out of this map? Please refer to the following:

Fuchs S., Ornetsmüller C., Totschnig R. 2012. Spatial scan statistics in vulnerability assessment: an application to mountain
hazards. Natural Hazards, 64: 2129-2151.

Fuchs et al. (2012) also detected spatial distribution patterns of loss ratios in four torrent fans in Austria.

**Answer on tables and figures:**

To avoid offline and unsynchronised modifications, several variable categories were included in the questionnaire
beforehand, without knowledge about the specific situation in the research area. We consider the presentation of the complete set of possible answers to be relevant for comparisons with followup studies. The table represents the complete
survey, as it was designed and the distribution of occurrences, categories with zero cases are thus included as well.

Due to the paper format, not all variables were included in the correlation tests. Our main objectives were to analyse the
damage driving factors of flash floods and to reveal potential differences compared to riverine flooding. E.g. Maiwald et al.
give an overview of known factors which influence structural damage on buildings. Especially the building material, condition (before the event) and the age are important factors related to the buildings resistance potential. Factors such as
inundation level and contamination relate to "action" parameters (Maiwald et al. 2015) and describe external forces. Thus,
the choice of the analysed variables was based on both, existing literature as well as expert judgement, i.e. including the
exposition in flow direction, the abundance of large shop windows on the ground level or the sealing of the near environment
as well. However, we analysed but not necessarily discussed all recorded variables in the revised paper.

Factors such as inundation level, flow velocity, specific energy and contamination relate to "action" parameters and describe
external forces (Maiwald and Schwarz, 2015). Thus, in our study, the inundation depth measured at the building and the
building's exposition in flow direction were combined to create a local impact, which can be seen as a proxy for local flood
related impact and hydrostatical forces at the building. Consequently, we chose a combination of these factors where both
contribute to equal extents. While the inundation depth has continuous values, which are roughly uniformly distributed between 2 and 360 cm, the exposition in flow direction is recorded in three classes (low, medium, high). To achieve
comparable variable ranges, the exposition classes "low", "medium" and "high" are transformed into the mean values of the
lower, middle and upper third of recorded water levels. The derived values 57, 133 and 230 fit into the range of observed
water levels, enabling a combination of both attributes (Figure 5). The calculated local impact corresponds to the sum of
water level and transformed exposition value. Please note that the exposition values are not used to replace water levels, but are only transformed into a comparable range.. In the revised paper, we support the explanation of this particular method
with a graphic.

The "process intensity"/"local impact"-map was created in QGIS and is used to visualize the "local impact" indicator.
However, it has to be noted that the local impact is measured on buildings itself and therefore hypothetical for the areas around. In the framework of the paper, the map is further used to illustrate the flash flood process in Braunsbach and to
underline the impact of water depth and exposition on the resulting damage. Overall the estimation of a local impact could
be used in strategic planning of mitigation measures against future hazards in Braunsbach. The same approach can also be
used for similar villages in that region, given that information about the potential flash flood (e.g. inundation depth) is
available either from observations of an actual event or from flash flood models. The consideration of exposition as damage
driver fits to the statement by Fuchs et al. (2012), saying that the general land use and settlement patterns play an important
role in the geographical distribution of building damage. Thus, our map may contribute to the identification of potentially
vulnerable locations on a small scale and within case studies.

**Referee #2**

**General answer:**

We thank the reviewer for the constructive comments. As stated by referee 1 as well, we agree with the reviewer that the Brief Communication had to be transformed into a Research paper, in which we extend our analysis and discussion.

**Comments of the reviewer 2**

**Reviewer2 quote, paragraph 1, 2, 3 & 4:**

The paper describes the field survey and some first results after the flash flood event of Braunsbach in Baden-Württemberg in Germany. This type of event and the analyses are an interesting and relevant topic in the field of building damage due to extreme flood events. The complex characteristic of these extreme flood events and the resulting, in some cases, very heavy structural damage is not only in Germany an insufficient understood problem.

The aims of the paper are the identification of the damage relevant parameter due to flash floods and a discussion about the benefits of the use of the open source software "KoBoCollect" for the data acquisition.

The paper gives a short overview about the process of the event and the investigation area. The relevant aspects of preparation and realization of the data collection during the field survey are described.

During the field survey, the authors classified the damaged buildings into a damage classification system developed by other authors. A damage grade as a measure for the structural damage was assigned to each damage case. These damage grades and the documented impact and building parameter are the basis for the statistical analyses for the identification of the damage-relevant parameters. These statistical analyses are a further focus of the paper. From the viewpoint of the referee, the linkages between the individual steps of the described procedure are logical and comprehensible.

**Answer 1, 2, 3 & 4:**

Thank you for acknowledging the relevance of our work.

In our research we used the damage classification scheme developed by Schwarz and Maiwald (2007) in order to ensure comparability to other studies. In the revised version, we keep the general outline and extended our research as well as literature review.

**Reviewer2 quote, paragraph 5 & 6:**

The principle problem of the paper is mentioned by the first referee. A "brief communication" should represent a significant contribution to science, ground breaking and new results…

In its present form the paper would be in principle a good damage report after correcting some inaccuracies. But in the present form it fulfils not the demand for a brief communication. In general there are two possibilities: to find a journal that accepts a report form or like suggested by the first referee, to extend the work to a research paper including a detailed analysis with more graphs and figures. In the latter case also more topic related literature should be cited. In each case the type of impact (flash flood, debris flow or mud flow) should be clearly separated with respect to the involved material components.

**Answer 5 & 6:**

Thank you for this suggestion. As also stated in our response to the comments of reviewer 1, we agree that the conversion of the Brief Communication into a Research paper was a helpful suggestion. We included more graphics supporting our methods and analysis. These are

1.  Graphics to explain our methods (i.e. the derivation of the process intensity/local impact in the revised paper)
2.  Graphics related to more detailed analysis

Our detailed analysis includes correlation tests with all recorded variables. We perform an additional analysis (i.e. multinomial logistic regression) to obtain additional outcomes and deepen the discussion. Further, we discuss our work in the context of existing studies on this topic.

The flash flood in Braunsbach was accompanied by a considerable amount of sediment and building rubble, potentially
showing flow characteristics of debris flows such as defined by Fuchs et al. 2010 and Borga et al 2014. Yet, a clear distinction between flash floods and debris flows is not always straightforward. In the revised version of the paper, we clearly define flash floods and debris flows and we aim for consistency and adequate wording for the process.

**Reviewer2 quote, paragraph 7:**

Some other comments are necessary: By the application of the damage classification system, the authors speak from the assignment of damage classes or degree of damage. In contrast, the original publication refers to the term "damage grades".

**Answer 7:**
Thank you for the comment, we have been more careful and consistent with the terms we use to describe the damage classification. In the revised version, we use the term "damage grades".

**Reviewer2 quote, paragraph 8:**
According the paper, the team was in the field first one week after the event. This is related with the careful preparations before the survey. However, it should be discussed whether the damages a week after the event still clearly assessable due to the advanced clean-up work. It could be also discussed, whether the water level measurements with the thermographic camera the ascending humidity in the walls was taken into account.

**Answer 8:**

One week after the event, the structural damage on buildings and building characteristics was still assessable, since the main work within this period was mainly focused on clearing the roads, establishing paths for large construction machinery as well as removing and cleaning the interior of affected buildings. Some areas of the village were even not accessible before, since roads were blocked by debris and buildings in danger of collapse had to be secured. The progress of the clean-up work was further beneficial for the damage assessment, as a thick layer of debris and rubble previously covered big parts of the building damage. However, few buildings could not be reliably examined, since debris and rubble were still hampering the access.

The use of the thermographic camera is better presented and discussed in the revised paper, since it offered advantages in such cases, where the inundation depth could not be reliably estimated. We agree with the reviewer, that the ascending humidity in the walls is a point to consider when using a thermographic camera for water level estimations. For that reason the thermal images were mainly used to verify estimations based on visible mud contamination and marks caused by water and transported debris. Since the thermally derived water levels matched well with visible traces, thermal images were also used to estimate the inundation depth for buildings with no or little visible traces.

**Reviewer2 quote, paragraph 9:**

A discussion about the topic process intensity seems also necessary. The first referee has here the opinion that the exposition belongs not to intensity. I believe at the end this is a question of the understanding of the meaning of intensity. Should the intensity considered only as a combination of impact parameter (water level, velocity, material density and debris impact)? Or has it an extended meaning like for earthquake according to EMS-98 (Grünthal et al. 1998), where also the effects on humans, nature and building were considered for the assignment of the intensity? Clear, for the damage also the exposition of the building can be relevant (Maiwald & Schwarz, 2015). A high exposition leads by such dynamic impact characteristics to higher loads on the buildings. With respect to these dynamic impacts especially the legitimation of the replacement of mean water level for some calculated percentiles with the exposition classes is unclear. Is there really a meaningful correlation?

**Answer 9:**

We agree with the statement of the reviewer that the attributes of intensity are determined by different meanings and can include exposition grades as well. In our case, the process intensity includes factors which are independent from building characteristics and do not only represent flood inherent parameters. As stated in the answer on the first referee comment as well, we replaced the term "process intensity" with "local impact", to reflect the actual meaning in a better way.

The exposition classes needed to be transformed in order to derive the process intensity. Still, the inundation depth and the exposition classes contribute independently to the process intensity. In the following paragraph, the methods are described in detail (copy from the answers on referee comment RC1):

"Factors such as inundation level, flow velocity, specific energy and contamination relate to "action" parameters and describe external forces (Maiwald and Schwarz, 2015). Thus, in our study, the inundation depth measured at the building and the building's exposition in flow direction were combined to create a local impact, which can be seen as a proxy for local flood related impact and hydrostatical forces at the building. Consequently, we chose a combination of these factors where both contribute to equal extents. While the inundation depth has continuous values, which are roughly uniformly distributed between 2 and 360 cm, the exposition in flow direction is recorded in three classes (low, medium, high). To achieve comparable variable ranges, the exposition classes "low", "medium" and "high" are transformed into the mean values of the lower, middle and upper third of recorded water levels. The derived values 57, 133 and 230 fit into the range of observed water levels, enabling a combination of both attributes (Figure 5). The calculated local impact corresponds to the sum of water level and transformed exposition value. Please note that the exposition values are not used to replace water levels, but are only transformed into a comparable range." In the revised paper, we support the explanation of this particular method with a graphic.

**Reviewer2 quote, paragraph 10:**

It could be not expected, that these complex topic can be analysed in a really detailed form from a limited study of 96 damage cases. Therefore is more comprehensive data base necessary. But after a major revision of this paper and its extension to a research paper we can expect more detailed insights in the topic. I look forward to the further progress of the work.

**Answer 10:**

We thank the reviewer and aim to contribute to a better understanding of flash floods and related damage processes in general. Considering the fact that our research in Braunsbach resembles a case study, the 96 damage cases represent an almost complete inquiry. Thus, we do not claim to perform an extensive study of this complex topic. However, we will add to the knowledge within this field of research. Especially since further research is planned and an additional survey related to flash floods will be carried out soon, connecting to our results.

**Summary of major paper changes:**

Since the paper format has changed from a "Brief Communication" to a full research paper, the changes in the text and graphics are very comprehensive and poorly to be shown by track changes of the former version. I therefore decided to list all major changes below:

- Included an elaborate literature review in the introduction.
- Added a new method to analyse the "local impact" indicator (multinomial logistic regression).
- Included new graphics (RF and RGLM model results, multinomial logistic regression, explanatory graphics).
- "local impact"-map update, change in the method of local impact "interpolation" (Voronoi diagrams instead of IDW interpolation were used).
- Elaborate discussion of significant variables, referring to literature.
- Less focus on open source software discussion and presentation of "KoBoCollect".

---

## Referee Report (RR1)

**REVIEWER'S REPORT**

I have now read the improved version of the research paper with the title „Damage assessment in Braunsbach 2016: A data collection and analysis for an improved understanding of damaging process during flash floods". The paper has been successfully transformed from a brief communication to a research paper and has been significantly improved, therefore, I believe that it should be considered for publication following some minor revisions. In particular, the following point should be considered:

1. The detailed description of the case study and the event in the introduction seems to me out of place. I would suggest to move the paragraph "Intense rainfall at the end of May (p.2, line 18) …need to catch up (p.3, line 5)" later in the text, probably in p. 8, at the beginning of the session: Results and discussion.
2. In the abstract the authors suggest that "the results reveal that the damage driving factors of flash floods differ from those of riverine floods to a certain extend". This suggestion should be discussed more in the text.
3. There is still an inconsistency regarding the number of variables. In p. 9, line 3 you are talking about 21 variables. Yet, the number of variables in Table 1 and Figure 3 is 18, whereas the number of variables in Figures 1 and 2 is 17. Try to be consistent.
4. I have the feeling that "conclusions" should be short, summing up the main results and outcome of the research. What is included now in "Conclusions" I would name "discussion" and move it to the previous chapter. I also think that it would be nice to include in the new "discussion" section a paragraph summing up assumptions and limitations such as "ascending humidity in the walls", "personal variations in expert judgement", etc.

---

## Referee Report (RR2)

The paper provides an interesting example of analysis of damaging processes (both with documentary use and future planning of mitigation measures in this and similar villages) and of application of useful software. I think the Authors thoroughly answered the questions of the previous reviewers and expanded the former Brief Communication into a paper. However, it still requires a few improvement by introducing some corrections as listed below.

In general, I think a flaw of the paper in its version of full Research Paper is the absence to references and discussion on the use of quantitative model based on the solution of the flow field during a flash flood.
Actually, in several paragraphs the Authors rightly observe that it is not only the water depth but also the exposition of a building to explain high damage, along with the features listed in Table 1. As shown by Figure 1 and 2, the two major components of the damage can be explained by depth and exposition. Contrary to the opinion of Reviewer 1 ("the exposition of the building in flow direction has to do with the orientation of the building itself and not with the process") I think that exposition has a meaning mostly because it is a proxy for the dynamic action exerted by flow on a building. Accordingly, thinking at a prognostic use of the methods presented in the paper, exposition could also be identified by computing forces on buildings impacted by flow. It seems to me that this is the meaning also of ther answer to Reviewer 2, paragraph 9.

In this direction, depth and velocity can be computed in an excellent way by numerical techniques based on the solution of De Saint Venant equations (DSVE) and these in turn can be used to compute exposure on a physical basis, by computing the specific force on a target as done in (Milanesi et al., 2015)

- Milanesi, L., Pilotti, M., Ranzi, R. (2015), A conceptual model of people's vulnerability to flood, Water Resources Research, 51, doi:10.1002/2014WR016172

Accordingly I think that, in order to provide a more comprehensive view of the problem, as required by a full Research Paper, the Authors should mention about this possibility in the Conclusions or in the Discussion. This conclusion could envisage a synergy between quantitative hydraulic approach and the methods presented in the paper. Actually the modeling approach provides only 2 of the parameters listed in Table 1.

| pag/line | Note |
| --- | --- |
| 30 | 6 km ? Is this the length of the main reach of the creek ? Probably the area is more relevant because more directly related to the peak flow. Please try to add a short outline of the drained watershed (area, maximum elevation, outlet elevation, average slope). A map of the watershed with shading representing elevation would be a plus. In a full paper a better description of the event could be well justified. |
| 31 | Estimated ? Revised ? Can you be less generic ? Where did you measure rainfall ? Why did you revise it ? How far from the watershed ? You can add the point on the map above. Is it possible to estimate the return period of the event? |
| Pag 5 | Is it possible to add any information about the classification of the damage classes? Maybe some pictures might help to differentiate more clearly. |
| Pag 6 | An explanation of the role of the extimated construction year would be useful |

| | |
|---|---|
| | since it reflects both the conservation of the building and the type of building technology. It would be interesting to understand which one of these aspects was most relevant in the examined watershed. |
| pag7/6 | With reference to Figures 1 and 2, in Paragraph 3.2 can you better explain the hierarchy adopted by the 2 methods because the orderings shown in Figure 1 and 2 are quite different, apart from the 2 first criteria (depth and exposure). |
| Page 8, line7 | Considering the relevant presence of sediment transport, one of the key parameters affecting the impact force on a obstacle is the estimated fluid density. (see also the paper mentioned above Milanesi et al., 2015) |
| | |
| Pag. 12 | The first sentence is affected by some confusion on the terminology. I would consider to take into account the fundamental definition of risk by Varnes (1984) and revise the entire paper in the light of such terms. In particular , considering that the local impact indicator is a proxy of the acting forces, it could be considered an expression of hazard. Accordingly, it is not necessary to compute exposure. Finally, vulnerability is an intrinsic property of each target (e.g. building, human, vehicle, etc.) that is not affected by the level of hazard or  exposure. The combination of such elements allows to compute risk, that is a representation of damage. |
| 20/5 | no to...: please correct typo |
| | Figures |
| pag25 | Add reference to Table 1 and section 2.3.2 in the caption for the meaning of local impact and of D1,D2,D3 |
| | |
| | References: |
| | the following papers are cited in the paper but not listed in the references: Molinari et al., 2014

is Ziese et al. (cited at pag. 2), 2017 or 2016 as in the References ?

is Murawski et al., 2016 or 2015 as in the References ?

is the following publication (see page 16) cited anywhere in the paper ?

Grüning, H., and Grimm, M.: Unwetter mit Rekordniederschlägen in Münster, KW Korrespondenz Wasserwirtschaft, 8, 2, 88-93, doi:10.3243/kwe2015.02.001, 2015 (in German). |
| Fig. 6 | It is not clear why in some heavily damaged buildings in the lower part of the domain the local impact factor is low. I would expect more correlation since the local impact is a function of depth and exposure, that are strongly correlated to damage. |

---

## Author Response (AR2)

**Author comments to:**

**Anonymous Referee 1: nhess-2016-387-referee-report-1**

**Anonymous Referee 2: nhess-2016-387-referee-report-2**

5  Author: Jonas Laudan[1]

Co-authors: Viktor Rözer[2], Tobias Sieg[1/2], Kristin Vogel[1], Annegret H. Thieken[1]

[1]University of Potsdam, Institute of Earth and Environmental Science, Karl-Liebknecht-Strasse 24-25, 14476 Potsdam, Germany

10  [2]GFZ German Research Centre for Geosciences, Department of Hydrology, Telegrafenberg, 14473 Potsdam, Germany

15  # Referee 1

**General answer:**

We thank the reviewer for the helpful and constructive comments and addressed all raised issues during the revision.

**Reviewer1 quote 1:**

20  The detailed description of the case study and the event in the introduction seems to me out of place. I would suggest to move the paragraph "Intense rainfall at the end of May (p.2, line 18) …need to catch up (p.3, line 5)" later in the text, probably in p. 8, at the beginning of the session: Results and discussion.

**Answer 1:**

25  Thank you for the suggestion, the paragraph was moved at the end of the introduction section (as suggested by the editor), where it fits better now in our opinion as well.

**Reviewer1 quote 2:**

In the abstract the authors suggest that "the results reveal that the damage driving factors of flash floods differ from those of riverine floods to a certain extend". This suggestion should be discussed more in the text.

**Answer 2:**

The differences of flash floods compared to riverine floods are now better discussed and presented in the introduction and in the discussion section.

**Reviewer1 quote 3:**

There is still an inconsistency regarding the number of variables. In p. 9, line 3 you are talking about 21 variables. Yet, the number of variables in Table 1 and Figure 3 is 18, whereas the number of variables in Figures 1 and 2 is 17. Try to be consistent.

**Answer 2:**

We rephrased particular sentences hoping to clarify the variable count inconsistency. Also in the figures 1 and 2 and section 3.2, we mentioned the use of the damage grade as response variable.

**Reviewer1 quote 4:**

I have the feeling that "conclusions" should be short, summing up the main results and outcome of the research. What is included now in "Conclusions" I would name "discussion" and move it to the previous chapter. I also think that it would be nice to include in the new "discussion" section a paragraph summing up assumptions and limitations such as "ascending humidity in the walls", "personal variations in expert judgement", etc.

**Answer 4:**

We renamed section 3.1: "Data collection and KoBoCollect" to "Data collection and field work; assumptions and limitations", since it better represents the discussed topics in that section. Further, we deepened the discussion regarding the ascending humidity in the walls. We think that this chapter now fulfils the demands of an "assumptions and limitations" paragraph, as suggested by the reviewer. Further, we did not move the content of the conclusion section, since we added a new chapter in the discussion (3.5), focusing on potential uses of a local impact indicator (see general answer to reviewer 2).

**Referee 2**

**General answer:**

Again, we thank the reviewer for the helpful and constructive comments and addressed all raised issues during the revision. In particular, we appreciate the suggestions and references concerning the computation of flow fields during flash floods. This is an interesting topic which we addressed in the new discussion section 3.5, but - as suggested by the editor - did not focus on in greater detail. However, in the new section 3.5 we focused on how to retrieve variables useful for a derivation of indicators such as the "local impact" and discussed prerequisites as well as potential limitations.

**Reviewer 2 quote 1:**

6 km ? Is this the length of the main reach of the creek ? Probably the area is more relevant because more directly related to the peak flow. Please try to add a short outline of the drained watershed (area, maximum elevation, outlet elevation, average slope). A map of the watershed with shading representing elevation would be a plus. In a full paper a better description of the event could be well justified.

**Answer 1:**

Additional to the size of the catchment area (which is 6 square kilometre) we added a more detailed description of the creek in the introduction section (length of the creek, slope) and rephrased parts of the paragraph. Since a paper has been submitted to Science of the Total Environment which extensively focuses on the hydrological and geological aspects of the event in Braunsbach (Ozturk, U., Wendi, D., Crisologo, I., Riemer, A., Agarwal, A., Vogel, K., López – Tarazón, J.A., Korup, O.: Rare flash floods and debris flows in southern Germany, submitted to: Sci. Total Environ., 2017.), we decided not to include maps and in-detail descriptions of the watershed. In the mentioned article, maps of the catchment area as well as elaborate geological and hydrological analyses are presented.

**Reviewer 2 quote 2:**

Estimated ? Revised ? Can you be less generic ? Where did you measure rainfall ? Why did you revise it ? How far from the watershed ? You can add the point on the map above. Is it possible to estimate the return period of the event?

**Answer 2:**

Thank you for the hints. We added a better description of the measured rainfall and rephrased the paragraph for a better understanding. As mentioned above, we did not explain the measurements and adopted methods in detail, since the focus of our paper lies on the data collection and post-hoc damage assessment. An extensive discussion and presentation of the

hydrological aspects can be found in the submitted article mentioned in Answer 1 as well as in the German paper by Bronstert et al. (2017).

5 **Reviewer 2 quote 3:**

Is it possible to add any information about the classification of the damage classes? Maybe some pictures might help to differentiate more clearly.

**Answer 3:**

10 Thank you for this very helpful comment. A graphic has been added which shows examples of damage grades.

**Reviewer 2 quote 4:**

An explanation of the role of the estimated construction year would be useful since it reflects both the conservation of the
15 building and the type of building technology. It would be interesting to understand which one of these aspects was most relevant in the examined watershed.

**Answer 4:**

We extended the description and effects of the estimated construction year in the discussion section 3.2. In retrospect, we
20 unfortunately do not have any detailed information on the importance of building aspects such as the state of technology and/or conservation for the respective damage.

**Reviewer 2 quote 5:**

25 With reference to Figures 1 and 2, in Paragraph 3.2 can you better explain the hierarchy adopted by the 2 methods because the orderings shown in Figure 1 and 2 are quite different, apart from the 2 first criteria (depth and exposure).

**Answer 5:**

Thank you for the suggestion. We agree that a deeper discussion on the variable importance is needed. In section 3.2, it is
30 now better explained why the analyses resulted in a different variable importance hierarchy in both models (RF and RGLM).

**Reviewer 2 quote 6:**

Considering the relevant presence of sediment transport, one of the key parameters affecting the impact force on a obstacle is the estimated fluid density. (see also the paper mentioned above Milanesi et al., 2015)

5 **Answer 6:**

In section 2.3.2, we mentioned the fluid density as a key parameter when explaining the external acting forces in case of flooding.

10 **Reviewer 2 quote 7:**

The first sentence is affected by some confusion on the terminology. I would consider to take into account the fundamental definition of risk by Varnes (1984) and revise the entire paper in the light of such terms. In particular, considering that the local impact indicator is a proxy of the acting forces, it could be considered an expression of hazard. Accordingly, it is not necessary to compute exposure. Finally, vulnerability is an intrinsic property of each target (e.g. building, human, vehicle, 15 etc.) that is not affected by the level of hazard or exposure. The combination of such elements allows to compute risk, that is a representation of damage.

**Answer 7:**

Thank you for the hint; we rephrased the sentence for a better understanding. We further added Varnes (1984) to the 20 references and adopted clearer definitions of risk in the new section 3.5.

**Reviewer 2 quote 8:**

No to...: please correct typo

**Answer 8:**

We rephrased the description of the damage grades in Table 1.

30 **Reviewer 2 quote 9:**

The following papers are cited in the paper but not listed in the references:

Molinari et al., 2014

is Ziese et al. (cited at pag. 2), 2017 or 2016 as in the References ?

is Murawski et al., 2016 or 2015 as in the References ?

is the following publication (see page 16) cited anywhere in the paper ?

Grüning, H., and Grimm, M.: Unwetter mit Rekordniederschlägen in Münster, KW Korrespondenz Wasserwirtschaft, 8, 2, 88-93, doi:10.3243/kwe2015.02.001, 2015 (in German).

5 **Answer 9:**

We corrected the referencing mistakes and added/deleted literature which was not cited properly or unused.

**Reviewer 2 quote 10:**

10 It is not clear why in some heavily damaged buildings in the lower part of the domain the local impact factor is low. I would expect more correlation since the local impact is a function of depth and exposure, that are strongly correlated to damage.

**Answer 10:**

Referring to one particular building in the lower parts of the town, it has to be mentioned that this building was old and the
15 condition before the event extraordinary weak. It had been found collapsed to a certain degree and was about to be demolished (we did not specifically mention this in the text). This example shows that, apart from the local impact on a building, certain aspects of vulnerability and building characteristics were influencing the damage as well, which is likewise supported by our results.

**List of Changes in the paper, according to all suggestions:**

We marked all changes in the document with a reference to the respective point in the list of changes.

5   Due to an elaborate database check in the last weeks and within the scope of the publication of the database, we changed definitions and adjusted values to assure database consistency regarding few variables (number of storeys & definition of different building materials). Based on taken pictures we discovered few inconsistencies during the checks which we corrected. Two data points had to be removed. This has a minor effect on model and correlation values and a very minor effect on the maps and figures. The overall results of our analysis and interpretation of these results however did not change.

10  This is referred to as point 1 in the list of changes.

1.  Checked inconsistent variables in the database and count of variables, clarification.
2.  Moved description of the case study in the text.
3.  Better description of the difference between riverine floods and flash floods, better focus.
15  4.  Changes in section 3.3: renamed section, merged sections 3.3 and 3.4, added a discussion on how information on significant variables for damage assessment can be retrieved.
5.  Damage grades, added figure for better explanation.
6.  Revised sentence regarding hazard & exposure.
7.  Renamed section 3.1: now section for assumptions and imitations, better discussion of some aspects.
20  8.  Clearer description of the event concerning rainfall.
9.  Explained the role of the variable: estimated construction year.
10. Better explanation: variable importance of Random Forest and Random Generalized Linear Model.
11. Added sentence regarding the role of fluid density and the impact force.
12. Clarified typo: clearer description.
25  13. Description of figure 4: added reference to table 2 (former table 1).
14. Checked overall references for mistakes and listing errors, added and removed literature.

[revised manuscript text omitted]